# Nanostructured Delivery Systems for Curcumin: Improving Bioavailability and Plaque-Targeting Efficacy in Atherosclerosis

**DOI:** 10.3390/pharmaceutics17111465

**Published:** 2025-11-13

**Authors:** Yu Liu, Tengfei Yu, Chao Zhang, Zhiyong Yang, Dahai Yu, Bin He, Yan Liang

**Affiliations:** 1International Education Department, Henan Vocational College of Nursing, Anyang 455000, China; c0013@hnnvc.edu.cn; 2Department of Pharmaceutics, School of Pharmacy, Qingdao University, Qingdao 266073, China; yutengfei60@gmail.com (T.Y.); ggbond1571@gmail.com (Z.Y.); 3Medical Psychology Center, Qingdao Special Servicemen Recuperation Center of PLA Navy, Qingdao 266021, China; zhangchao20251112@gmail.com; 4National Engineering Research Center for Biomaterials, Sichuan University, Chengdu 610064, China; bhe@scu.edu.cn

**Keywords:** atherosclerosis, curcumin, inflammation, nanoformulations, drug delivery

## Abstract

Cardiovascular disease (CVD) encompasses ischemic conditions of the heart, brain, and bodily tissues, primarily resulting from hyperlipidemia, atherosclerosis (AS), hypertension, and other related factors. CVD accounts for over 40% of global non-communicable disease mortality, making it the leading cause of death and a significant medical burden worldwide. AS, the principal pathological basis for most cardiovascular diseases, is characterized as a chronic, sterile inflammatory condition triggered by lipid overload and various other factors. In recent years, natural bioactive compounds have gained prominence in the treatment of human diseases. Among these, curcumin (Cur) has garnered considerable attention due to its anti-inflammatory, lipid-lowering, antihypertensive, and endothelial protective properties. This review examines traditional pharmacological approaches for treating AS, with particular emphasis on the critical mechanisms through which Cur exerts its therapeutic effects. Additionally, it introduces novel nanoformulations designed to address the inherent limitations of Cur, providing valuable insights for researchers investigating its application in AS therapy.

## 1. Introduction

CVD encompasses ischemic or hemorrhagic disorders affecting the heart, brain, and systemic tissues due to factors such as hyperlipidemia, blood viscosity, AS, and hypertension [1,2]. Globally, cardiovascular diseases are the primary cause of mortality and medical issues, accounting for over 40% of non-communicable disease deaths worldwide [3,4]. This condition is characterized by significant morbidity, disability, and mortality rates [5]. Given the widespread prevalence of lifestyle risk factors like unhealthy diet choices, inadequate physical activity levels, and smoking habits, there has been a continuous rise in the absolute number of individuals affected by hypertension, dyslipidemia, diabetes, and obesity in China [6,7,8,9]. Consequently, this trend is expected to further contribute to an increase in both incidence and mortality rates associated with cardiovascular disease. It is estimated that there are currently approximately 330 million cardiovascular patients in China, with cardiovascular disease (CVD) surpassing tumors as the leading cause of mortality [10,11]. CVD in China is currently experiencing a “double climb” trend for both prevalence and mortality. In 2020, CVD accounted for 48.00% and 45.86% of rural and urban deaths, respectively, making it the primary cause of death in both areas. Globally, two out of every five deaths are attributed to CVD, highlighting its significance as a major public health issue worldwide [12]. AS is the predominant pathological basis for most cardiovascular diseases, characterized by chronic sterile inflammation induced by lipid deposition and foam cell (FC) formation, among other factors. While traditional lipid-lowering and anti-inflammatory drugs have demonstrated remarkable therapeutic effects against AS, they are also associated with various side effects, such as gastrointestinal dysfunction and potential liver or kidney damage [13]. In recent years, research on natural active ingredients has deepened, leading to increased attention to natural plants and their effective components due to their minimal side effects, acceptability, and ability to regulate the structure and function of intestinal flora. Active ingredients derived from natural sources are widely used in the treatment of human diseases [14]. Cur, berberine, resveratrol, and other drugs have been progressively employed for the prevention and treatment of AS. Cur is a polyphenol compound extracted from the rhizomes of Cur longa, Curcuma longa, and Curcuma zedoary, with a chemical formula of (E,E)-1,7-bis(4-hydroxy-3-methoxyphenyl)-1,6-heptadiene-3,5-dione. It possesses various biological activities, such as anti-inflammatory properties, lipid-lowering effects, blood pressure-lowering effects, and endothelial cell protection [15]. Research indicates that Cur plays a significant role in preventing and treating AS, along with its associated complications [16]. However, Cur exhibits low bioavailability, which restricts its clinical application [17]. With the rapid advancements in the nanotechnology field, Cur has been formulated into diverse new nano-preparations to further enhance its stability and bioavailability. This article provides an overview of traditional drugs used for AS treatment over the past few years, emphasizing the crucial role of Cur in this context. Additionally, it focuses on novel nanopreparations designed to address its inherent limitations, thereby offering valuable insights for researchers interested in using Cur to treat AS.

## 2. Formation and Development of AS

AS primarily occurs in the subintimal layer of large and medium arteries with a diameter larger than 3 mm. Common arteries affected by AS include the coronary, carotid, aorta, and renal arteries. Previously considered a degenerative disease associated with aging, recent research suggests that AS development is predominantly driven by an intricate interplay between inflammation and lipids. Its pathogenesis involves various cell types, including monocytes, endothelial cells, vascular smooth muscle cells, monocyte-derived macrophages, dendritic cells, and regulatory T cells [18]. The main manifestations include arterial endothelial injury, lipid deposition leading to thickening and hardening of the arterial wall, loss of elasticity, luminal stenosis, and cellular death and fibrosis [19]. The development of AS is primarily attributed to disturbances in lipid metabolism and vascular dysfunction, resulting from the synergistic effects of multiple factors. The key contributors to AS include hyperlipidemia, hypertension, diabetes, obesity, smoking, and fatigue [20,21].

As shown in Figure 1, the pathological progression of AS can typically be categorized into four stages: early formation of fatty streaks, development of fibrous plaques, formation of atherosclerotic plaques, and occurrence of complex lesions or secondary changes (such as intraplaque hemorrhage, plaque rupture, and thrombosis). Following endothelial cell dysfunction, low-density lipoprotein (LDL) enters the vascular intima through damaged endothelium and undergoes oxidation, forming oxidized low-density lipoprotein (Ox-LDL). Ox-LDL activates vascular endothelial cells, inducing them to express chemokines and adhesion factors that recruit monocytes from the bloodstream. Monocytes adhere to activated vascular endothelial cells, migrate toward the inner wall of blood vessels, and subsequently differentiate into macrophages. Scavenger receptors on macrophage surfaces rapidly recognize and engulf Ox-LDL. Following the ingestion of a substantial amount of Ox-LDL, cholesterol ester-containing lipid droplets form in the cytoplasm, disrupting the normal balance of lipid metabolism within the cells and impairing macrophage migration from blood vessels to generate FCs. FCs, cholesterol, fibrocytes, and smooth muscle cells, among others, gradually accumulate to form atherosclerotic plaques. As the disease progresses, the phagocytic function of macrophages diminishes, leading to the death of FCs and the subsequent disintegration of cell membranes. This results in the release of lipids and the formation of necrotic cores within plaques. In advanced stages of AS, excessive lipid accumulation along with inflammatory reactions ultimately trigger increased activity levels of matrix metalloproteinases (MMPs) in the plaque microenvironment, particularly MMP-2 and MMP-9, which can degrade the extracellular matrix, causing severe erosion of the fibrous cap and continuous expansion of the necrotic core. Consequently, stable plaques transform into vulnerable ones that are prone to rupture, hemorrhage, and thrombosis, thereby obstructing blood flow to tissues [22,23]. This leads to organ ischemia and hypoxia, resulting in acute cardiovascular and cerebrovascular events associated with a significantly high disability rate as well as mortality. The initial stage of AS development progresses at an exceedingly slow pace, posing challenges for detection. Subsequently, plaque formation leads to stenosis of the blood lumen, resulting in reduced or even obstructed blood flow. Furthermore, when plaque dislodges, it readily triggers thrombus formation, thereby causing blockage in downstream blood vessels and giving rise to severe consequences such as myocardial infarction and stroke.

## 3. Traditional Medicine for AS

Effectively controlling and treating atherosclerosis is paramount for reducing mortality from cardiovascular and cerebrovascular diseases. It also plays a significant role in enhancing individuals’ happiness, quality of life, and overall lifespan. AS can be prevented through dietary adjustments and lifestyle improvements, such as increasing the consumption of whole grains, legumes, vegetables, and fruits; reducing saturated fat and cholesterol intake; engaging in regular physical exercise; and regularly monitoring blood lipid levels [27,28]. The treatment of AS is categorized into surgical intervention and pharmacological therapy. Surgical procedures such as revascularization, reconstruction, or bypass grafting can be performed on narrowed or blocked arteries. Interventional treatments, such as placing stents in blood vessels, can also be performed. However, due to drawbacks associated with surgical procedures, including arterial vessel rupture risk, tissue detachment complications, thrombosis-inducing potential leading to accelerated formation of new atherosclerotic plaques, and postoperative complications, they may fail to address the fundamental issue of AS recurrence [29]. Henceforth, drug therapy is more commonly employed for treating AS. Pharmacotherapeutic options primarily include lipid-lowering agents, anti-inflammatory drugs, antiplatelet medications, and anticoagulants [30], as shown in Figure 2. In recent years, novel therapeutic strategies targeting oxidative stress, inflammation, and immune regulation have emerged, including mTOR inhibitors [31,32,33] (e.g., rapamycin), nanomedicine-based delivery systems [34,35], and anti-inflammatory biologics (e.g., IL-1β or TNF-α inhibitors). These approaches aim to stabilize plaques, promote cholesterol efflux, and restore vascular homeostasis, offering new directions for treating atherosclerosis.

### 3.1. Hypolipidemic Drug

Abnormal lipid metabolism is a crucial pathogenic factor in the development of AS, exerting a significant impact on the progression of coronary artery disease. Consequently, lipid-lowering therapy has consistently remained at the forefront of both preventing and treating AS [30]. Statins are inhibitors of hydroxymethylglutaryl-CoA reductase, effectively impeding mevalonate formation and inhibiting endogenous cholesterol synthesis [36,37]. Simultaneously, reduced cholesterol concentrations lead to an upregulation in LDL receptor numbers and activity on cell membranes, facilitating increased absorption of LDL particles and subsequently lowering plasma LDL and overall cholesterol levels.

Simultaneously, statins inhibit vascular smooth muscle cell migration, reduce intravascular inflammation, improve plaque stability, and exert other effects that effectively prevent the progression of AS. Statins are widely used in primary and secondary prevention and the treatment of cardiovascular diseases; however, they have side effects such as myotoxicity and elevated transaminases [38,39]. Commonly used statins in clinical practice include atorvastatin, rosuvastatin, and simvastatin. Fibrates are phenoxyaromatic acid lipid-regulating drugs that activate the peroxisome proliferator-activated receptor (PPARa) and induce lipoprotein esterase expression to promote triglyceride hydrolysis in triglyceride-rich lipoprotein particles, leading to a reduction in plasma very low-density lipoprotein (VLDL) levels. PPARa activation enhances key genes involved in high-density lipoprotein cholesterol (HDL-C) metabolism and promotes HDL-C synthesis. Simultaneously, increased expression of HDL receptors accelerates cholesterol efflux from extrahepatic cells and uptake by hepatocytes, thereby promoting reverse cholesterol transport. Fibrates such as fenofibrate, gemfibrozil, and bezafibrate are commonly used clinically to prevent and treat AS. The lipid-lowering mechanism of niacin remains unclear; however, its effect on reducing VLDL synthesis and distribution in the liver may be related to its inhibition of lipolysis in adipose tissue [40].

### 3.2. Anti-Inflammatory Drugs

Inflammation is a prevalent clinical pathological process that can manifest in various tissues and organs throughout the body. The crux of the inflammatory response lies in the generation and targeting of inflammatory factors. To date, key signaling pathways have been identified, including the MAPK-dependent, TLR-dependent, ROS-dependent, and phosphatidylinositol 3-kinase-protein kinase B (PI3K-Akt) pathways [39,41]. Research has demonstrated that inflammation occurring in both adventitia and intima plays a pivotal role in the initiation and progression of AS. AS is the consequence of chronic and excessive inflammation of the vascular wall. The theory of inflammation is one among several theories explaining the etiology of AS. Inflammation increases endothelial cell permeability, thereby increasing the likelihood of lipid infiltration and accelerating atherosclerotic progression. The inflammatory process in atherogenesis involves both innate immune cells (such as macrophages and monocytes) and adaptive immune elements (including B cells and T cells), along with various cytokine cascades.

Anti-inflammatory medications have the potential to impact the inflammation/thrombus interface, thereby improving and treating AS [42]. Anti-inflammatory medications have been shown to effectively inhibit the development of AS. Other anti-inflammatory drugs, such as Cur, methotrexate, tocilizumab, etanercept, and Canakinumab, developed for different targets, are currently showing promise for treating AS [43,44].

### 3.3. Antiplatelet Drugs

Antithrombotic therapy is widely used for atherosclerotic thrombosis, in which platelets actively participate in the formation and tissue remodeling of AS, playing pivotal roles in hemostasis and mediating atherosclerotic plaque rupture and thrombosis. Aspirin effectively inhibits platelet adhesion and aggregation by suppressing cyclooxygenase synthesis, thereby reducing prostaglandin production and impeding platelet thromboxane A2 generation [45]. It is the most commonly employed medication for both preventing and treating AS. Proteasome inhibitors exert notable effects on platelet function and hemostasis when specifically targeted; bortezomib (Velcade), an inhibitor, induces thrombocytopenia, leading to fatality. Following platelet activation, dense granules release ADP and thromboxane A2 via thromboxane synthase activation, perpetuating ADP-P2Y12 receptor signaling and glycoprotein IIb/IIIa receptor activation, which ultimately results in platelet aggregation and subsequent thrombus formation. The P2Y12 receptor antagonist (clopidogrel) enhances its inhibitory effect on coronary artery thrombosis while concurrently reducing the incidence of AS [46].

### 3.4. Anticoagulants

Anticoagulant medications inhibit thrombin production, activate plasminogen, and dissolve fibrin to achieve anticoagulation. Rivaroxaban is an oral anticoagulant that selectively inhibits factor Xa, thereby preventing blood clotting by suppressing thrombin production. It is used to prevent and treat venous thromboembolism as well as prevent systemic embolism in stroke or atrial fibrillation [47]. Heparin is a natural anticoagulant substance present in the animal body, characterized by its polymer structure composed of alternating polysaccharide chains. It exhibits enhanced affinity for both antithrombin and thrombin, thereby accelerating thrombin inactivation [48]. Additionally, it augments protein C activity and stimulates vascular endothelial cells to release anticoagulant and fibrinolytic substances. With demonstrated efficacy both in vivo and in vitro, heparin is a widely used clinical anticoagulant that plays a crucial role in the treatment of AS. Oral administration of heparin proves ineffective; however, intravenous injection, intravenous drip, or deep subcutaneous injection can be employed for therapeutic purposes [49].

Although statins and other traditional drugs have proven effective in reducing the morbidity and mortality of cardiovascular disease, their long-term use is still limited by side effects and tolerability, such as liver toxicity and myotoxicity of statins. In addition, these drugs often target only a single pathological pathway, while atherosclerosis is a chronic inflammatory disease involving multiple factors such as lipid disorders, oxidative stress, and immune inflammation. In contrast, natural compounds such as curcumin can regulate multiple pathological links of AS simultaneously due to their multi-target mechanism of action. However, its poor water solubility and rapid metabolism lead to low bioavailability, thus limiting its clinical application. In order to overcome these shortcomings, in recent years, nanotechnology-based delivery systems have become an effective means to improve the stability, absorption and targeting of curcumin. These nanoparticles can not only increase the drug concentration of curcumin in atherosclerotic lesions but also achieve controlled and sustained release. In summary, the combination of natural drugs such as curcumin with nano-drug delivery systems, which combine the multi-target advantages of natural compounds with the precision and efficiency of modern nano-delivery technology, provides new opportunities for long-term and safe cardiovascular treatment.

## 4. Cur in the Treatment of AS

Turmeric is a dual-use material for medicine and food. It is the dried rhizome of turmeric, a member of the ginger family. Turmeric is listed as a natural food additive by the Food and Drug Administration of the United Nations World Health Organization. It can be used as a seasoning and food coloring, being one of the main spices used in curry. Its main effects include activating blood circulation, warming the meridians, soothing the liver, regulating qi, dredging channels, relieving pain, and so on. It has high medicinal value and is a common medicinal material for treating cardiovascular diseases in traditional Chinese medicine [50]. Its rhizomes for medicinal use are mainly produced in Fujian, Guangdong, Guangxi, and Yunnan. To date, more than 200 compounds have been identified in turmeric, including essential oils (4.2~14%), fatty oils (4.4~12.7%), and other volatile oils. Cur is the most important active ingredient in turmeric. Curcumin is an orange crystalline powder classified as a diketone compound, comprising two *o*-methylated phenolic rings linked by a β-diketone chain. Its molecular structure is defined as 1,7-bis(4-hydroxy-3-methoxyphenyl)-1,6-heptadiene-3,5-dione [51,52]. Traditional clinical drugs, such as statins and anti-inflammatory drugs, have clear efficacy in reducing blood lipid levels and preventing cardiovascular events. However, their long-term use may cause side effects such as liver damage, muscle disorders, and drug resistance. In contrast, curcumin, a natural polyphenolic compound, offers multiple benefits—including anti-inflammatory, antioxidant, lipid-regulating, and endothelial-protective effects—and is relatively safer.

Curcumin was prioritized over other bioactive compounds (e.g., resveratrol and quercetin) due to its unique multi-targeted mechanisms against the pathogenesis of atherosclerosis. It simultaneously inhibits inflammation (e.g., a 40–60% reduction in TNF-α/IL-6 via NF-κB/MAPK suppression in murine models) [53], reduces oxidative stress (ROS-scavenging capacity (IC_50_ = 12.5 μM) [54], and improves lipid metabolism (a 35% increase in ABCA1-mediated cholesterol efflux in THP-1 macrophages) [55]—which single-pathway compounds rarely replicate. Its superior translational potential is supported by clinical evidence: a 2022 randomized trial (*n* = 120) showed a 0.18 mm reduction in carotid intima-media thickness (CIMT) after 6 months of 1500 mg/day of curcumin–phospholipid complex [56], whereas resveratrol (500 mg/day) yielded only a 0.07 mm CIMT reduction in similar cohorts [57]. Pharmacokinetically, modern formulations (e.g., solid lipid nanoparticles) enhance curcumin’s bioavailability by 20–30-fold (Cₘₐₓ = 2.3 μM, T_1_/_2_ = 8.5 h) compared to unformulated curcumin (Cₘₐₓ < 0.1 μM) [58], enabling therapeutic concentrations in atherosclerotic plaques (measured at 1.8 μM in ApoE^−/−^ mice) [59]. Focusing on curcumin ensures mechanistic depth, enabling detailed investigation of its interactions with key targets (e.g., endothelial NO synthase and macrophage CD36), rather than diluting the findings through broad comparative analysis.

Cur has been extensively researched in recent years, and its pharmacological effects have been confirmed to include anti-tumor, anti-inflammatory, antioxidant, hypoglycemic, lipid-lowering, and analgesic properties [50,60].

It has wide clinical applications in the treatment of diseases such as tumors, obesity, and diabetes. With advancements in molecular pharmacology and biology, among other cutting-edge disciplines, the medicinal potential of Cur has been further explored through extensive studies focusing on its role in treating AS (Figure 3). Its mechanism primarily revolves around the aspects elaborated upon in the following section [61].

### 4.1. Inhibition of the Inflammatory Response

The initial focus of AS formation studies was primarily on lipid metabolism. However, as research has advanced, more recent theories have identified inflammation as a crucial risk factor and key mediator of arterial wall cells. Inflammation plays a pivotal role in the pathological changes in the vascular endothelium, closely linked to the expression of inflammatory cytokines. Inflammatory reactions are evident at various stages of AS development [62]. Cur’s anti-inflammatory properties underpin its diverse biological activities and play a crucial role in disease treatment. It can regulate various inflammatory signaling pathways, including TNF-α, IL-1β, IL-6, transforming growth factor-β (TGF-β), ROS, and Ox-LDL, thereby inhibiting vascular inflammation. Its regulatory mechanisms include binding to the Toll-like receptor (TLR) [63,64,65], inhibiting the NF-κ B pathway, suppressing mitogen-activated protein kinase MAPK phosphorylation (ERK, p38MAPK, and JNK) and Janus kinase (JAK/STAT) activation, inhibiting the PI3K-Akt signaling pathway, and blocking the CD40–CD40 L signaling pathway [66]. These cytokines normally promote endothelial activation, upregulate adhesion molecules (VCAM-1 and ICAM-1), and facilitate monocyte adhesion and infiltration into the vascular intima—key early steps in plaque formation. By suppressing IL-12 and interferon-γ (IFN-γ), Cur inhibits the Th1-type immune response, whereas enhancing IL-10 and TGF-β promotes an anti-inflammatory and plaque-stabilizing environment. These regulatory effects contribute to macrophage polarization toward the M2 phenotype, favoring tissue repair and the resolution of inflammation [65].

Cur has the ability to regulate macrophage phenotypes and inhibit the body’s inflammatory response. The immune cells present in atherosclerotic plaques include T cells, B cells, NK cells, NKT cells, macrophages, monocytes, dendritic cells, neutrophils, and mast cells. During inflammation, circulating monocytes migrate from the peripheral blood to the subintima, where they encounter a microenvironment rich in growth factors and pro-inflammatory factors that induce their differentiation into macrophages. Macrophages are the predominant immune cell type within atherosclerotic plaques and play a crucial role in the development of AS. Macrophage phenotypes can be classified into M1 (classical activation) and M2 (alternative activation) phenotypes. M1 macrophages promote inflammation and secrete pro-inflammatory factors, such as IL-1β and IL-6, whereas M2 macrophages inhibit inflammation and stimulate the production of anti-inflammatory factors, such as IL-10. In atherosclerotic plaques, M1 macrophages are predominant, leading to increased levels of IL-6 and IL-1β, which exacerbate atherosclerotic lesions. The primary approach to suppressing the inflammatory response is to modulate macrophage polarization toward the M2 phenotype or reduce the prevalence of the M1 phenotype [67,68]. Studies have shown that promoting the polarization of macrophages toward the M2 type reduces the development of AS in ApoE^−/−^ and Reversa mice. Cur can attenuate the release of inflammatory cytokines and decrease the presence of M1 macrophages by inhibiting NF-κB activation and downregulating the overexpression of DNA methyltransferase 3b (DNMT3b). Additionally, Cur significantly inhibits the TLR4/MAPK/NF-κB signaling pathway while enhancing STAT-6 phosphorylation, thereby promoting macrophage transformation from a pro-inflammatory to an anti-inflammatory phenotype. Furthermore, Cur activates PPARγ to promote macrophage polarization from the M1 to the M2 phenotype [69]. Since AS is driven by inflammation, with distinct stages involving distinct roles for macrophages in its pathogenesis, curcumin’s ability to promote macrophage transition from M1 to M2 has become a key focus for future research [70].

Cur also reduces atherosclerotic inflammation by inducing macrophage autophagy [71]. In Figure 4, Ox-LDL accumulates in the atherosclerotic plaque and is phagocytized by macrophages. Excessive lipid accumulation transforms these macrophages into FCs, which exhibit significant autophagy deficiencies. This deficiency leads to the secretion of pro-inflammatory factors and exacerbates lipid deposition, thereby accelerating the formation of a necrotic core in atherosclerotic plaques. The metabolic abnormalities in FCs primarily result from LDL’s impact on the transcription factor EB (TFEB)-P300-bromodomain-containing protein 4 (BRD4) axis. Cur effectively restores autophagy in FCs by promoting TFEB nuclear translocation, optimizing lipid catabolism, reducing inflammation, and mitigating inflammatory responses [71,72].

### 4.2. ROS Scavenging and Antioxidant

Oxidative stress is a harmful condition caused by the presence of free radicals in the body and is widely recognized as a major contributor to aging and disease progression. They are key initiators in the development of AS [73,74] and can directly damage vascular endothelial cells and increase the expression of intercellular adhesion molecule-1, vascular cell adhesion molecule-1, E-selectin, and P-selectin. This process promotes immune cell adhesion to endothelial surfaces and their subsequent migration into subcutaneous tissues, worsening vascular inflammation. Additionally, ROS can oxidize LDL in vivo into Ox-LDL, which is more pathogenic. This oxidation damages endothelial cells, increases their permeability, and widens the gaps between them, allowing more cholesterol, LDL particles, and inflammatory factors to infiltrate subcutaneous layers [75,76]. This leads to increased lipid deposition and inflammation, creating a vicious cycle. Ox-LDL is taken up by macrophages, leading to lipid accumulation and the formation of FCs [77]. Therefore, reducing systemic oxidative stress, preventing vascular oxidative stress, and decreasing ROS production within plaques are viable strategies for treating AS. Cur, a widely used antioxidant with diverse medicinal properties, exerts its effects primarily by directly eliminating ROS. Stimulated cells generate ROS that disrupt cellular redox balance, damaging DNA and proteins, compromising cell structure and function, and contributing to disease. ROS include superoxide radicals, hydroxyl radicals, hydrogen peroxide, and singlet oxygen molecules, which are highly oxidizing agents that react with reducing substances. Antioxidants inhibit oxidation reactions and protect normal cellular functions. Antioxidants such as ergothionein, vitamin C, vitamin E, glutathione (GSH), melatonin, α-lipoic acid, and carotenoids have ROS-scavenging properties. Many antioxidants contain phenolic hydroxyl groups, which are compounds in which one or more hydrogen atoms on a benzene ring are replaced by hydroxyl groups. This feature is common in natural products like polyphenols and flavonoids, contributing to their potent antioxidant capacity [78,79]. The antioxidative mechanism of phenolic hydroxyl groups involves several processes: they can react with singlet oxygen to convert it into stable molecular oxygen, decompose hydrogen peroxide into water and oxygen, neutralize free radicals generated through lipid peroxidation, and form chelates with metal ions (such as copper or iron ions), inhibiting their catalytic effect on oxidation reactions. Cur contains a phenolic hydroxyl structure with reducing properties, enabling it to effectively eliminate ROS and act as an antioxidant, thereby protecting the vascular endothelium and improving AS [80].

Cur exerts indirect antioxidant effects by inducing the synthesis of cytoprotective proteins. GSH, a tripeptide composed of glutamic acid, cysteine, and glycine, possesses antioxidant and detoxification properties. Cur activates the Keap1-Nrf2-ARE signaling pathway, leading to the dissociation of the Nrf2-Keap1 complex in the cytoplasm. This process facilitates the nuclear translocation of the transcription factor Nrf2, where it binds to the antioxidant response element (ARE) and activates transcription of downstream target genes. Additionally, Cur enhances the activity of glutamate cysteine ligase (GCL), a rate-limiting enzyme involved in GSH synthesis, and stimulates the cystine-glutamate reverse transporter to facilitate the cellular uptake of cystine [81]. This provides cysteine, a precursor necessary for GSH synthesis, thereby promoting GSH production. Heme oxygenase-1 (HO-1) is a key antioxidant enzyme that protects cells against both endogenous and exogenous stimuli. As a heat shock protein, HO-1 can be directly activated to resist oxidative stress caused by various stimuli. Cur can promote HMOX1 synthesis, thereby reducing oxidative stress in cells. Superoxide dismutases (SODs) are antioxidative metalloenzymes that catalyze the dismutation of superoxide radicals to oxygen and hydrogen peroxide [82]. They play an important role in maintaining the balance between oxidation and antioxidation. Curcumin can increase superoxide dismutase activity, accelerate ROS clearance, and protect cells [83].

### 4.3. Hypolipidemic

Blood lipid indicators typically include total cholesterol, triglycerides, HDL-C, and LDL-C. The development of AS is closely linked to lipid metabolism, with changes in lipid levels serving as a primary trigger. The accumulation of lipids beneath the blood vessel intima forms the foundation of AS. Different components of blood lipids have varying effects on the treatment of AS. Research has shown that HDL possesses anti-atherosclerotic properties, such as promoting cholesterol transport, exhibiting anti-inflammatory activity, and providing antioxidant effects [84]. Conversely, elevated levels of LDL and cholesterol contribute to lipid deposition and FCs formation, thereby exacerbating AS. Cholesterol levels play a pivotal role in the development of AS and are regulated through physiological processes, including synthesis, transportation, absorption, and excretion. Cur significantly reduces plasma levels of total cholesterol (TC), triglycerides (TG), and LDL-C while increasing HDL-C levels. Additionally, Cur effectively inhibits cholesterol absorption in the small intestine. Cholesterol uptake from dietary sources is facilitated by the Niemann–Pick C1-like 1 (NPC1L1) transporter located in the small intestine. The NPC1L1 gene is upregulated by the transcription factor SREBP-2. Inhibition or reduction in NPC1L1 activity leads to decreased cholesterol absorption and subsequently lowers blood cholesterol levels [85,86]. A study by Jun et al. demonstrated that curcumin reduces intestinal cholesterol absorption and plasma cholesterol levels in rats fed a high-fat diet as well as in ApoE^−/−^ mice [87].

Cur suppresses SREBP-2 transcription, thereby inhibiting the SREBP-2-NPCL1L1 signaling pathway and downregulating NPC1L1 expression in intestinal epithelial cells, ultimately reducing cholesterol absorption by these cells. Additionally, Cur decreases cholesterol synthesis in the liver, the primary site of this process. Shin et al. [88] demonstrated that Cur can inhibit HMG-CoA reductase activity and downregulate HMGR gene expression, resulting in reduced cholesterol synthesis. Cur has the ability to enhance reverse transport of lipids, inhibit FCs formation, and prevent lipid deposition. Liver X receptors (LXRs) are ligand-activated transcription factors belonging to the nuclear receptor superfamily that can detect changes in intracellular cholesterol levels. Cur is capable of activating AMPK, inducing phosphorylation of AMPK, increasing mRNA and protein expression of LXRα, promoting activation of LXRα and its binding with genes involved in cholesterol transport and metabolism, upregulating mRNA expression of genes related to cholesterol transport and metabolism such as ATP binding cassette (ABC) transporters ABCA1 and ABCG1, and enhancing ABCA1 expression on macrophage membrane, thereby facilitating cholesterol efflux [89]. Cur enhances fatty acid oxidation, a metabolic process in which fatty acids are broken down into CO_2_ and H_2_O under aerobic conditions, releasing significant energy [90]. It achieves this by upregulating hormone-sensitive lipases, acetyl-CoA carboxylase, carnitine acyltransferase, and peroxisome proliferator-activated receptor gamma coactivator-1 to stimulate lipolysis and induce fatty acid beta-oxidation. This mechanism is associated with increased lipid breakdown in mitochondria [91,92].

### 4.4. Lowering Blood Pressure

The presence of hypertension can lead to enhanced blood flow and increased impact on the arterial wall, resulting in heightened tension and mechanical damage. These conditions facilitate the development of atherosclerotic plaques. Additionally, hypertension can elevate peripheral resistance and sustain vasoconstriction, leading to inadequate blood supply to vital organs, such as the liver and kidneys. This compromised blood flow negatively affects lipid metabolism, further exacerbating AS [93]. Blood pressure regulation involves a multitude of neurohumoral regulatory factors, among which the renin–angiotensin–aldosterone system plays a pivotal role. In response to decreased blood pressure, the glomerular granular cells of the glomerular apparatus (also known as the glomerular complex) secrete renin, an enzyme also known as angiotensinogen. Renin acts on plasma angiotensinogen to generate angiotensin I, which lacks blood pressure-raising activity. Angiotensin I is subsequently hydrolyzed by angiotensin converting enzyme into angiotensin II, which exhibits hypertensive properties. Angiotensin II activates the angiotensin type 1 receptor (AT1R), thereby initiating activation of the renin–angiotensin–aldosterone system, vasoconstriction, increased myocardial contraction, and stimulation of adrenaline and aldosterone synthesis and release, ultimately leading to a significant elevation in blood pressure. AT1R blockers antagonize the binding of angiotensin II to AT1R receptors, resulting in reduced vasodilation and diminished renin and aldosterone release; they also decrease central and peripheral sympathetic nerve activity while lowering blood pressure.

Cur can inhibit AT1R expression by suppressing AT1R gene transcription, thereby attenuating the hypertensive effect induced by angiotensin binding to AT1R [94]. Angiotensin-converting enzyme (ACE), also known as kininase II or peptidyl-carboxypeptidase, converts angiotensin I, which lacks hypertensive effects, into angiotensin II with hypertensive effects, thus elevating blood pressure through activation of the renin–angiotensin–aldosterone system. ACE also deactivates bradykinin, inhibiting its vasodilatory properties and contributing to increased blood pressure. Cur can suppress ACE mRNA expression and activity, thereby impeding angiotensin II synthesis and reducing blood pressure [95]. Dysregulation of gut microbiota and gut–brain communication can also contribute to elevated blood pressure. Li et al. discovered that Cur is capable of reshaping intestinal biota composition and improving intestinal environmental homeostasis while rectifying gut–brain communication disorders via modulation of butyric acid-G protein-coupled receptor 43 (GPR43) pathway function in order to enhance autonomous cardiovascular regulation within the brain region [96].

### 4.5. Other Properties of Cur

Cur exhibits antithrombotic properties [97]. Imbalances in the coagulation and anticoagulation systems can promote the progression of AS and lead to thrombotic diseases. Thrombus formation is closely associated with platelet aggregation. Cur modulates thromboxane synthesis and inhibits platelet aggregation induced by arachidonic acid, adrenaline, and collagen, thereby exerting anticoagulant and antithrombotic effects [98,99]. Additionally, Cur possesses hypoglycemic properties [100]. In addition to diabetes, elevated blood sugar can also accelerate the development of AS. Elevated blood sugar can lead to abnormal lipid metabolism, damage vascular endothelial cells, and cause vascular rupture and bleeding, as well as induce expansion of the necrotic core within atherosclerotic plaques and calcification of arterial walls, thereby accelerating the progression of atherosclerotic lesions. Insulin is the sole hormone responsible for lowering blood sugar levels. The increase in blood sugar is closely associated with insulin resistance. Cur significantly reduces fasting blood glucose (FBG) and glycosylated hemoglobin A1c (HbA1c) levels, as well as the insulin resistance index (HOMA-IR), thereby improving insulin resistance, enhancing insulin sensitivity, and optimizing its ability to lower blood sugar [101,102]. Additionally, Cur enhances the electrical activity of islet cells and promotes insulin release, further contributing to its blood sugar-lowering effects. Moreover, Cur exhibits antiviral properties by inhibiting the hemagglutinin of H1N1 and H6N1 subtypes of influenza viruses, thereby directly impacting viral replication. It also disrupts the integrity of viral envelopes and liposome membranes, mitigating cardiopulmonary damage caused by the virus. Studies have demonstrated that in vivo administration of Cur at a dose range of 25–100 mg/kg reduces lung pathology in influenza-infected mice while decreasing their lung index value. Furthermore, it significantly prolongs average survival days and reduces mortality rates among virus-infected mice [103,104]. In addition to the aforementioned effects, Cur also exhibits anti-infective properties and provides hepatoprotective and nephroprotective benefits, thereby holding significant importance in the prevention and treatment of AS [105].

### 4.6. Clinical Research

One such trial aimed to evaluate the efficacy of curcumin on various cardiovascular risk factors in patients with coronary artery disease (CAD). A total of 33 patients with CAD, who met the inclusion and exclusion criteria, were enrolled in this study. The participants were randomly assigned to receive either curcumin (500 mg capsules, four times daily) or a placebo for 8 weeks. Lipid profile, blood glucose, and high-sensitivity C-reactive protein (hs-CRP) levels were measured at baseline and after two months of treatment. The results indicated that curcumin significantly reduced serum triglyceride, LDL, and VLDL levels compared with the baseline. However, there were no significant changes in total cholesterol, HDL, blood glucose, or hs-CRP levels. Although curcumin improved some lipid parameters, its effect on inflammatory markers was not significant in this small-scale study, suggesting that larger, longer-term clinical trials are needed.

This section highlights the pharmacological effects and molecular mechanisms of curcumin in atherosclerosis. Curcumin exerts a comprehensive multi-target therapeutic effect by simultaneously regulating lipid metabolism, oxidative stress, inflammation, and endothelial dysfunction. At the molecular level, it inhibits inflammatory signaling pathways and suppresses the production of pro-inflammatory cytokines, thereby preventing endothelial activation and leukocyte adhesion. Meanwhile, curcumin activates antioxidant enzymes to mitigate oxidative stress in vascular tissues. In addition, it reduces foam cell formation and lipid accumulation within plaques by inhibiting cholesterol synthesis and upregulating the transporters ABCA1 and ABCG1 to promote reverse cholesterol transport. Overall, through its synergistic multi-level actions, curcumin provides protection across multiple stages of atherosclerosis—from early endothelial injury to late plaque instability—making it a promising natural therapeutic agent.

## 5. Nanostructured Delivery Systems for Cur

Cur, derived from abundant plant sources, exhibits diverse pharmacological effects and holds promising application prospects. However, its clinical development and application are hindered by challenges such as poor solubility and stability, low bioavailability, high dosage requirements, and a lack of targeted delivery upon entering the body. Fortunately, with the rapid advancement of nanotechnology, nanomaterials have found extensive applications in various fields, including materials science, the food industry, electronics engineering, textile manufacturing, chemistry, and biomedicine [106,107].

Nanodrug carriers are solid colloidal particles ranging in size from 1 to 100 nm, composed of natural or synthetic polymer materials. Common nanodrug delivery systems can be categorized into organic nanodrug delivery systems based on organic molecules, such as polymer micelles, liposomes, microcapsules, and microspheres, and inorganic nanodrug delivery systems based on inorganic elements, such as carbon nanotubes, iron oxide nanoparticles, mesoporous silica nanoparticles, and gold nanoparticles [108,109]. The size range of nanodrug carriers is comparable to that of proteins and other macromolecules in living cells, enabling them to selectively and efficiently penetrate cell membranes and tissue barriers. Due to the disparity in size between nanomaterials and macro-materials, nanomaterials can be artificially modified to alter their dimensions, shapes, surface charges, stabilities, and other characteristics. This allows them to possess a range of functionalities that conventional drugs lack, such as improved drug bioavailability, enhanced drug targeting and stability, conditional response capabilities, and increased cellular internalization. Furthermore, the nanodrug delivery system offers safety advantages, including reduced non-target cytotoxicity risk, a high biological safety profile, easy metabolism, and degradation. Given the high risks associated with surgery and the numerous adverse reactions caused by traditional anti-inflammatory drugs, the utilization of nanodrug delivery systems for AS has emerged as a highly promising treatment approach in current research [110]. The utilization of Cur-containing nanodrug delivery systems in the treatment of AS has garnered significant attention and research efforts, yielding the subsequent findings.

### 5.1. Polymeric Micelles

The self-assembly of amphiphilic copolymers results in the formation of polymer micelles (PMs), which serve as nanodrug carriers. These PMs feature a hydrophobic core that can encapsulate hydrophobic drugs or other bioactive substances, while their hydrophilic shell provides essential colloidal stability [111,112,113]. As shown in Figure 5, Ning et al. developed a dual-responsive drug delivery carrier based on poly(N,N-dimethylaminoethyl methacrylate) (PDMAEMA) as a pH-sensitive segment and poly(o-nitrobenzyl acrylate) (PNBAE) as a photoresponsive segment. The amphiphilic copolymer PEG–PDMAEMA–PNBAE was synthesized via atom transfer radical polymerization (ATRP) and employed for curcumin (Cur) delivery. In vitro drug release studies revealed cumulative release rates of 31.79% at pH 5.5 and 46.55% at pH 5.5 under UV irradiation, indicating efficient pH and light responsiveness [114]. Man et al. [115] linked Cur to amphiphilic methoxy poly (ethylene glycol)-poly (lactic acid) (mPEG-PLA) copolymers via acetal linkages to form polymeric micelles capable of releasing drugs in response to low-pH conditions, thereby efficiently delivering drugs into cells to exert their pharmacological effects. Siyu et al. [116] prepared Cur-loaded galactopolyglactic acid/TPGS micelles (CUR@GPP micelles) via the thin film dispersion method. The micelles were about 100 nm in diameter. Among them, GPP3 micelles with high-density Gal modification could obviously promote the uptake and absorption of Cur by intestinal epithelial cells through specific binding to the cellular galactopolyglactin receptor, thereby improving the bioavailability of Cur. This study highlights the significance of Gal content in the design of targeted nanocarriers modified with Gal for micelles, which have great potential for the oral delivery of hydrophobic drugs. Xiaoya et al. [117] prepared a novel amphiphilic carrier material, HASF, targeting the CD44 receptor and ROS sensitivity, and combined Cur and HASF into HASF@Cur micelles via self-assembly. The results showed that HASF@Cur responded to high concentrations of H_2_O_2_, releasing Cur at rates that increased with the H_2_O_2_ concentration, and targeting CD44 receptors in Raw 264.7 cells greatly enhanced Cur release at atherosclerotic plaque sites.

The above studies collectively demonstrate that PMs, owing to their tunable physicochemical properties and responsiveness to pathological microenvironments, offer an efficient and versatile platform for curcumin delivery. Moreover, through rational design, PMs can be engineered to achieve dual responsiveness to pH and light stimuli, while specific targeting of distinct atherosclerotic sites enables localized drug accumulation at lesion regions. These findings indicate that polymeric micelles not only enhance the solubility and bioavailability of curcumin but also enable spatially and temporally controlled drug release, underscoring their great potential as intelligent nanocarriers for targeted therapy of atherosclerosis and other inflammation-related diseases.

### 5.2. Liposome

Liposomes are ultra-fine spherical carrier systems composed of a hydrophilic core surrounded by a lipid bilayer. Thanks to their hydrophilic and hydrophobic properties, liposomes can deliver hydrophobic, hydrophilic, and amphiphilic drugs. Their biomembrane-like structure ensures low immunogenicity and excellent biocompatibility. Additionally, the lipid bilayer of liposomes provides drug protection, extends drug retention time in the bloodstream, reduces liver and kidney metabolism, and decreases drug toxicity [119,120]. As shown in Figure 6, the liposomes designed in this study encapsulate curcumin modified with hyaluronic acid (HA-Cur-LPs), allowing them to specifically recognize and bind to CD44 receptors overexpressed on cell surfaces due to hyaluronic acid modification. This facilitates the targeted delivery of curcumin. HA-Cur-LPs demonstrate enhanced stability, increased cellular uptake, and improved targeting compared to free curcumin and non-targeted liposomes (Cur-LPs) [121]. Jingxuan et al. [119] prepared a macrophage cell membrane-coated with Cur and celecoxib biomimetic liposomes modified with a cell-penetrating TAT-NBD fusion peptide (TN), denoted as PM/TN-CCLP. The results demonstrate that PM/TN-CCLP exhibits enhanced stability, improved biosafety, and a higher macrophage uptake rate than the free drugs. Moreover, it significantly inhibits macrophage transendothelial migration, reduces the cellular inflammatory response, and suppresses ROS production. Furthermore, PM/TN-CCLP enables co-delivery of Cur and celecoxib for synergistic anti-inflammatory effects. Xiaoxia et al. [122]. prepared E-selectin-binding peptide-modified liposomes, T-AC-Lipo, to co-deliver atorvastatin calcium and Cur to dysfunctional endothelial cells that overexpress E-selectin. The results demonstrated that T-AC-Lipo exhibits targeted drug delivery to atherosclerotic plaque sites and effectively disrupts the overexpression of adhesion molecules (E-selectin and ICAM-1) on endothelial cells. This disruption leads to a significant reduction in the expression of adhesion and inflammatory factors at plaque sites, decreased blood lipid levels, and notable attenuation of atherosclerotic lesions. As shown in Figure 6C,D, T-AC-Lipo demonstrates reduced cytotoxicity compared to free drugs while exhibiting significantly enhanced efficacy and safety. By encapsulating atorvastatin and Cur within liposomes and targeting them for simultaneous delivery to dysfunctional endothelial cells, a significant reduction in atherosclerotic lesions can be achieved with fewer side effects. This provides an excellent strategy for the synergistic anti-AS effect of drugs and targeted drug delivery. Ana Catalan-Latorre et al. [121] combined phospholipids, Eudragit^®^S100, and sodium hyaluronate salts to obtain Eudragit-hyaluronate immobilized vesicles using a simple and environmentally friendly method. For the first time, these two polymers were combined in a system for oral drug delivery to enhance the stability of Cur in the harsh GI tract environment. In vitro studies have shown that the combined polymer is able to protect the vesicles from harsh conditions of the GI tract, namely, changes in ionic strength and pH, a greater degree of Cur deposition in the intestinal region compared to the free drug in dispersion, and vesicles can ensure local protection from oxidative stress and increase their intestinal absorption.

The lipid bilayer structure of liposomes can effectively encapsulate and protect curcumin, thereby enhancing its stability and bioavailability. The customized functional surface-modified liposome system can achieve precise delivery to the lesion site, thereby enhancing the therapeutic effect. Additionally, the addition of biopolymers such as Eudragit^®^S100 and sodium hyaluronic acid can enhance gastrointestinal stability and promote intestinal absorption, thus enabling oral administration. In summary, these research results indicate that liposomes are a highly adaptable carrier that can integrate functions such as targeting, synergy, and protection. As a multifunctional nanocarrier for curcumin delivery, they have significant potential advantages in the treatment of atherosclerosis.

### 5.3. Nanoparticles

The presence of hypertension can lead to enhanced blood flow and increased impact on the arterial wall, resulting in heightened tension and mechanical damage. These conditions facilitate the development of atherosclerotic plaques. Additionally, hypertension can elevate peripheral resistance and sustain vasoconstriction, leading to inadequate blood supply to vital organs such as the liver and kidneys. This compromised blood flow negatively affects lipid metabolism, further exacerbating AS [123,124]. Liu et al. [125] reported a cholesteryl-9-carboxynonanoate–(^125^I iron oxide nanoparticle/curcumin)–lipid-coated nanoparticle (MLNP) system, in which the incorporation of “eat-me” signals (PtdSer and 9-CCN) within the lipid membrane effectively promoted macrophage internalization. This design enabled efficient delivery of ^125^I-IONs and curcumin to macrophages, resulting in a pronounced polarization shift from the pro-inflammatory M1 phenotype toward the anti-inflammatory M2 phenotype. Rasmita prepared CS-PLGA-AT-CU nanoparticles with chitosan-modified polylactic acid-co-glycolic acid as the shell and AT-CU, a conjugate synthesized by Cur and atorvastatin, encapsulated inside [126]. It was found that the PLGA shell of nanoparticles had good biocompatibility and could be degraded in vivo, but it burst easily. The problem of burst release could be solved by surface modification with chitosan. With increasing chitosan, the particle size of the nanoparticles increased from 139.2 nm to 197.7 nm, and the drug encapsulation efficiency increased from 71.81% to 90.57%. The safety and delivery efficiency of drugs were improved, which benefited AS treatment. As shown in Figure 7, Weidong et al. [127] designed a mesoporous MnO_2_ nanoparticle Cur-MnO_2_/HA with MnO_2_ prepared using the mixing/ultrasonic method as the core, surface modified with hyaluronic acid, and internally loaded with Cur. Hyaluronic acid modification can not only increase the stability of nanoparticles but also recognize and bind the CD44 receptor, which is overexpressed by macrophages at the atherosclerotic disease site, thereby enabling targeted delivery of Cur. MnO_2_ exhibits catalase-like activity, catalyzing the decomposition of H_2_O_2_ into oxygen, scavenging oxidative free radicals, and alleviating hypoxia at the plaque site. In addition, the mesoporous structure and the abundant metal coordination sites of MnO_2_ are beneficial for Cur loading. The results demonstrated that Cur-MnO_2_/HA exhibited a drug loading capacity of up to 54%, enhanced the pharmacokinetic performance of Cur, significantly suppressed the generation of ROS and cytokines, inhibited FCs formation, facilitated macrophage polarization toward the M2 phenotype, and displayed superior efficacy in the treatment of AS compared to free drugs, thereby exhibiting promising prospects for clinical application. During the development and progression of atherosclerosis (AS), oxidative stress in macrophages and endothelial cells plays a pivotal role in amplifying inflammation and promoting plaque formation. However, the lipid-rich core beneath the vascular intima often impedes the penetration of large-sized nanoparticles into the plaque, thereby limiting their access to dysfunctional macrophages. This barrier impedes the disruption of the vicious cycle of oxidative stress and inflammation between macrophages and endothelial cells. This work developed a biocompatible nano-drug loaded with curcumin, which can be a promising candidate drug for AS treatment. The constructed MnO_2_/HA has inherent catalytic activity and excellent drug-loading capacity. Given the extensive therapeutic and diagnostic applications of manganese-based nanomaterials, this nanoparticle has great potential for clinical translation. To address this challenge, Li et al. [128] developed a size-tunable Cur-based nanodrug delivery system. The platform was constructed using a Cur–human serum albumin (Cur–HSA) complex as the core, modified with polyarginine (R9) as a cell-penetrating peptide, and further coated with chondroitin sulfate (CS) as a targeting ligand, forming CS/R9@Cur–HSA nanoparticles. In vitro studies demonstrated that CS/R9@Cur–HSA exhibited enhanced macrophage-targeting capability, inhibited polarization toward the pro-inflammatory phenotype, and reduced ox-LDL-induced foam cell formation. Meanwhile, it improved the anti-inflammatory response and restored endothelial cell function. In vivo evaluations further confirmed that this nanoplatform effectively reduced plaque burden, lipid deposition, and inflammatory cell infiltration by synergistically regulating macrophage and endothelial cell oxidative stress, thereby delaying disease progression. This study provides a novel strategy for the application of size-tunable nanodrug delivery systems in the treatment of atherosclerosis.

In conclusion, these findings indicate that the multifunctional nanoparticles based on curcumin, with their antioxidant, anti-inflammatory, and cell-targeting properties, provide a comprehensive and adaptable treatment approach for hypertension-related atherosclerosis and its complications.

### 5.4. Microspheres and Microcapsules

Microspheres and microcapsules composed of natural or synthetic polymeric materials are advanced drug delivery systems (DDSs) capable of encapsulating therapeutic agents in solid or liquid form. Microcapsules typically store drugs within a core reservoir enclosed by a polymeric shell, whereas microspheres are spherical particles in which the drug is uniformly dissolved or dispersed throughout the polymer matrix. Due to their high surface area, resistance to enzymatic degradation, enhanced peptide stability, controlled release behavior, and site-specific targeting capabilities, microspheres and microcapsules have attracted considerable attention in biomedical applications. As an emerging DDS technology, drug-loaded microspheres overcome many limitations of conventional delivery methods by enabling sustained release, precise targeting, and broad drug compatibility, making them a versatile and highly promising platform in modern medicine [129,130]. Digambara et al. [131] developed spherical microcapsules encapsulating curcumin using poly-L-lysine, trisodium citrate, and silica sol as raw materials. The encapsulation efficiency of curcumin was increased to 57.34%, thereby enhancing its delivery efficacy. These microcapsules exhibited pH-responsive behavior and demonstrated high stability under alkaline or neutral conditions, with minimal drug release. In contrast, maximum drug release occurred under acidic conditions. Saranya et al. [132] prepared Cur-chitosan microspheres CCCM by first forming a curcumin–chitosan complex through a Schiff base reaction, followed by the solvent evaporation method to obtain the microspheres. The results indicate that these microspheres exhibit good biocompatibility, do not cause hemolysis, and possess strong anti-inflammatory and antioxidant activities, as well as a certain degree of antibacterial activity, suggesting promising medical application prospects.

### 5.5. Hydrogel

A hydrogel is a biodegradable and biocompatible matrix characterized by a hydrophilic three-dimensional polymer lattice, formed by the linkage of monomers or polymers via hydrogen bonds, electrostatic interactions, hydrophobic interactions, or covalent bonds. In recent years, hydrogels have emerged as highly promising polymers with significant potential for development and extensive biomedical applications [133]. Yuan et al. [126] successfully synthesized a PU-Cur modified polyurethane hydrogel by directly incorporating curcumin into a polyurethane prepolymer and crosslinking it with PCL-triol. Curcumin can be either physically blended or chemically bonded into polyurethane networks, serving as cross-linking agents and chain extenders. This hydrogel enhances curcumin’s stability, retains its antioxidant and anti-inflammatory properties, and prolongs its release time, offering significant application value. Recently, Zhang et al. [134] developed a composite hydrogel carrier that integrates a γ-cyclodextrin metal–organic framework (γ-CD-MOF) with β-cyclodextrin nanosponges (β-CDNS) to address the poor water solubility of Cur. The system achieved a high drug-loading capacity (13.9%) and a remarkable 267-fold increase in aqueous solubility. Due to the biointeractive properties and limited water absorption of β-CDNS, the γ-CD-MOF@β-CDNS composite hydrogel markedly enhanced the in vitro release and transdermal penetration of Cur. Wang et al. [135] developed a myocardial injury-responsive hydrogel formed by a reversible borate ester bond between carboxymethyl cellulose–boric acid (CMC-BA) and polyvinyl alcohol (PVA). Further, PLGA nanoparticles loaded with curcumin (PLGA@Cur NPs) and recombinant human collagen III (rhCol III) were incorporated into this network. The resulting hydrogel significantly promoted tissue regeneration and repair in the myocardial infarction area and improved cardiac function.

The above studies collectively demonstrate that the hydrogel-based system has high flexibility and therapeutic potential in the delivery of curcumin and tissue repair. Its hydrophilic three-dimensional network structure provides a favorable microenvironment for the sustained release of drugs and local treatment, highlighting its significant application potential in the treatment of cardiovascular diseases.

### 5.6. Nanosuspension

Nanocrystal suspensions (NSs), characterized by their submicron particle size and unique physicochemical properties, offer a versatile strategy for enhancing the delivery of drugs with poor water or lipid solubility. By significantly increasing the saturation solubility and dissolution rate of active pharmaceutical ingredients, NSs can markedly improve oral bioavailability and promote cell-mediated endocytosis. In addition, they reduce interindividual pharmacokinetic variability and minimize differences in drug absorption between the fasted and fed states. Due to their simple preparation process, high drug-loading capacity, and minimal excipient requirements, NSs have emerged as one of the most promising formulation strategies to enhance the biopharmaceutical performance of poorly water-soluble drugs, particularly natural products [136,137]. Lei et al. [138] prepared Cur nanosuspension Cur-ns using polyvinyl pyrrolidone (PVPK30) and sodium dodecyl sulfate (SDS) as stabilizers. The results demonstrated that Cur-ns was more readily taken up by RAW 264.7 cells compared to free curcumin. Additionally, under ultrasound stimulation, there was increased accumulation of Cur-ns in cells, which promoted macrophage apoptosis and facilitated macrophage transformation from the M1 to the M2 phenotype. This transformation helps inhibit the progression of AS.

### 5.7. Other Cur Delivery Platforms

In addition to nanoparticle and hydrogel systems, various new delivery platforms have been developed in recent years to enhance the efficacy and stability of curcumin in the treatment of cardiovascular diseases. Among them, electrospun nanofibers have attracted much attention due to their ability to achieve local and sustained drug release and their structural and mechanical properties that mimic the natural extracellular matrix (ECM). Electrospun fibers are usually prepared from degradable polymers such as polycaprolactone (PCL), polyvinyl alcohol (PVA), and chitosan, and can serve as bioactive scaffolds. Studies have shown that PCL or PVA electrospun nanofibers loaded with curcumin can alleviate oxidative stress and reduce the expression of inflammatory factors, thereby promoting myocardial tissue repair and endothelial cell proliferation, and have a positive therapeutic effect on ischemic heart damage [139]. The researchers integrated magnetic nanoparticles into the fiber structure and proposed a new intelligent drug delivery platform based on electrospun magnetic fibers (EMFs). They co-assembled drugs, magnetic nanoparticles, and mesoporous silica nanoparticles in the same fiber matrix. By applying an alternating magnetic field, the drug release process could be precisely controlled. The research results showed that this system had excellent encapsulation efficiency and universality for hydrophobic drugs such as Cur (Table 1) [140].

## 6. The Limitations of Curcumin in Clinical Applications

Although curcumin has shown potential cardiovascular benefits in some preclinical studies and small-scale clinical trials, the existing research still has several limitations. First, most clinical studies on curcumin have small sample sizes and short intervention periods, limiting the generalizability of their results. Additionally, there is an inconsistency in the effects of curcumin on inflammatory markers and atherosclerotic plaques, which may be related to differences in research design, curcumin dosage, and assessment methods. Therefore, large-scale, long-term randomized controlled trials are needed to further verify the therapeutic potential of curcumin in cardiovascular diseases such as atherosclerosis. Another significant challenge is curcumin’s low bioavailability. Curcumin has poor water solubility, a low absorption rate, and is rapidly metabolized and excreted, which limits its clinical efficacy. Therefore, researchers are increasingly using nanotechnology to improve the bioavailability and therapeutic effect of curcumin. Nanoparticle drug delivery systems are designed to increase the solubility and stability of curcumin and achieve targeted delivery to specific tissues, especially atherosclerotic plaques.

Although these nanotechnology-based delivery systems have great potential, they also face challenges. Designing stable and biocompatible nanoparticles requires precise control of multiple factors, such as particle size, surface charge, and release kinetics, placing high demands on the development of technologies. Moreover, the long-term safety of nanomaterials, especially their potential toxicity, immunogenicity, and accumulation in organs such as the liver and spleen, remains a significant concern. Therefore, further research is needed to evaluate the safety and stability of nanocarriers for clinical applications, enabling their widespread use in curcumin-based treatments for atherosclerosis and other cardiovascular diseases. Future research on our artificial curcumin delivery system for the treatment of atherosclerosis will focus on improving the targeting of the drug to atherosclerotic plaques, the controllable release in the inflammatory microenvironment, and reducing systemic toxicity while enhancing therapeutic efficacy. Intelligent nanocarriers that can respond to plaque-specific stimuli (such as acidic pH, ROS, or enzyme activity) are expected to achieve precise, local release of curcumin.

## 7. Conclusions and Perspectives

AS is a complex and chronic inflammatory cardiovascular disease involving multiple pathological processes, which remains one of the leading causes of cardiovascular events and mortality worldwide. Curcumin, a natural polyphenolic compound, has attracted considerable attention due to its potent anti-inflammatory, antioxidant, lipid-regulatory, and endothelial-protective properties, making it a promising natural therapeutic candidate for preventing and treating AS (as shown in Figure 8). Extensive in vitro studies have demonstrated that curcumin can inhibit key inflammatory signaling pathways (e.g., those involving NF-κB and NLRP3 inflammasome activation), reduce the release of pro-inflammatory cytokines, promote cholesterol efflux from macrophage foam cells, alleviate lipid accumulation, and improve endothelial function. Despite these encouraging preclinical findings, the clinical translation of curcumin remains challenging. Its poor aqueous solubility, rapid metabolism, and fast systemic clearance lead to extremely low bioavailability, thereby limiting its pharmacological efficacy in vivo. In recent years, advances in pharmaceutics and nanotechnology have offered new strategies to overcome these limitations. Various curcumin delivery systems have been developed to enhance its pharmacokinetic properties and targeting. For example, lipid-based formulations, polymeric micelles, and stimuli-responsive or ligand-modified nanocarriers have been shown to significantly improve curcumin’s absorption, circulation time, and stability, enabling sustained release and targeted delivery. These emerging delivery platforms create new possibilities for the precise delivery of curcumin to atherosclerotic lesions. Although substantial progress has been achieved in formulation technology, several key challenges remain before curcumin can be successfully translated into clinical use. Future research should focus on optimizing formulation design and large-scale manufacturing processes, comprehensively evaluating long-term pharmacokinetics and toxicological safety, and establishing standardized quality control systems to ensure batch-to-batch consistency. Moreover, well-designed, large-scale randomized clinical trials are urgently needed to confirm the efficacy and safety of curcumin-based formulations in patients with cardiovascular disease. Integrating nanotechnology, pharmaceutical engineering, and clinical pharmacology is crucial to further enhance the stability, bioavailability, and therapeutic efficacy of curcumin formulations. Multidisciplinary collaboration will help to bridge the gap between basic research and clinical application, ultimately promoting curcumin and its innovative formulations as promising strategies for preventing and treating cardiovascular diseases.

## Figures and Tables

**Figure 1 pharmaceutics-17-01465-f001:**
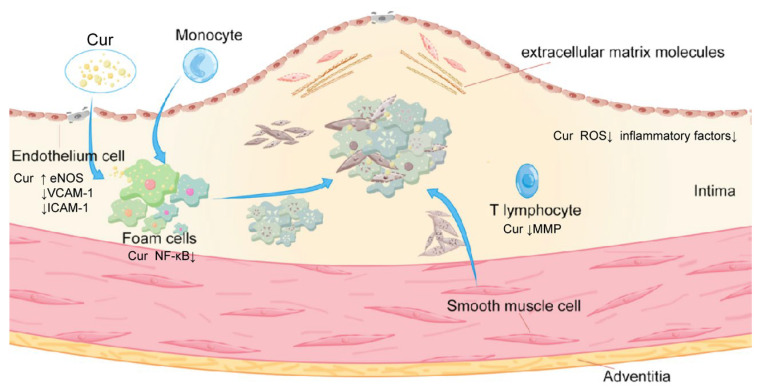
A schematic diagram of the progression of atherosclerosis and the intervention effect of curcumin. Endothelial retention of LDL cholesterol particles and damage to the monolayer of arterial endothelial cells increase permeability and further retain LDL particles. Endothelial cells are activated to express chemokines and cytokines, leading to the recruitment of monocytes, which differentiate into macrophages after ingesting LDL containing apolipoprotein B (Apo B). Macrophages undergo lipid overload, transform into FCs, and smooth muscle cells migrate and proliferate, synthesizing extracellular matrix macromolecules which are conducive to the formation of fiber caps. With the progression of AS, hypoxia at the lesion site leads to apoptosis, and lipid accumulation in apoptotic cells forms a lipid core. Cur inhibits endothelial activation and maintains barrier integrity by suppressing NF-κB-mediated inflammatory signaling, thereby reducing the expression of adhesion molecules (VCAM-1, ICAM-1) and chemokines (MCP-1) [24,25]. Cur also prevents LDL oxidation and scavenges reactive oxygen species, limiting ox-LDL-induced endothelial injury and foam cell formation. In macrophages, Cur suppresses lipid uptake by downregulating CD36 and SR-A while promoting cholesterol efflux via ABCA1 and ABCG1, thereby attenuating foam cell accumulation [26]. These integrated effects collectively slow AS progression and enhance plaque stability. Figure 1 was created by the authors.

**Figure 2 pharmaceutics-17-01465-f002:**
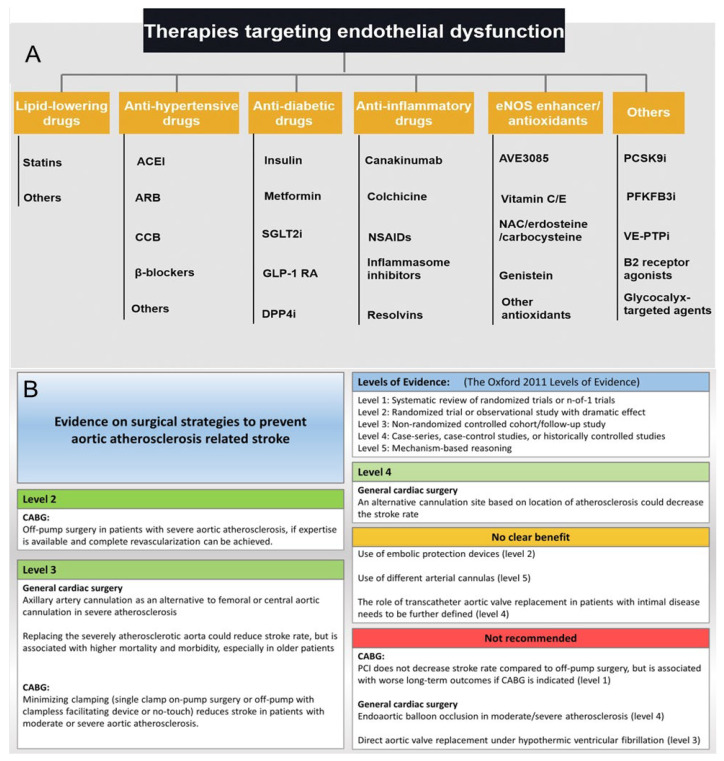
(**A**) Different drug treatments for endothelial dysfunction [30]. Reproduced from Ref. [30] with permission from Elsevier Ltd., (Amsterdam, The Netherlands) copyright 2021. (**B**) A visual summary of assessment strategies [29]. Reproduced from Ref. [29] with permission from Elsevier Ltd., copyright 2021.

**Figure 3 pharmaceutics-17-01465-f003:**
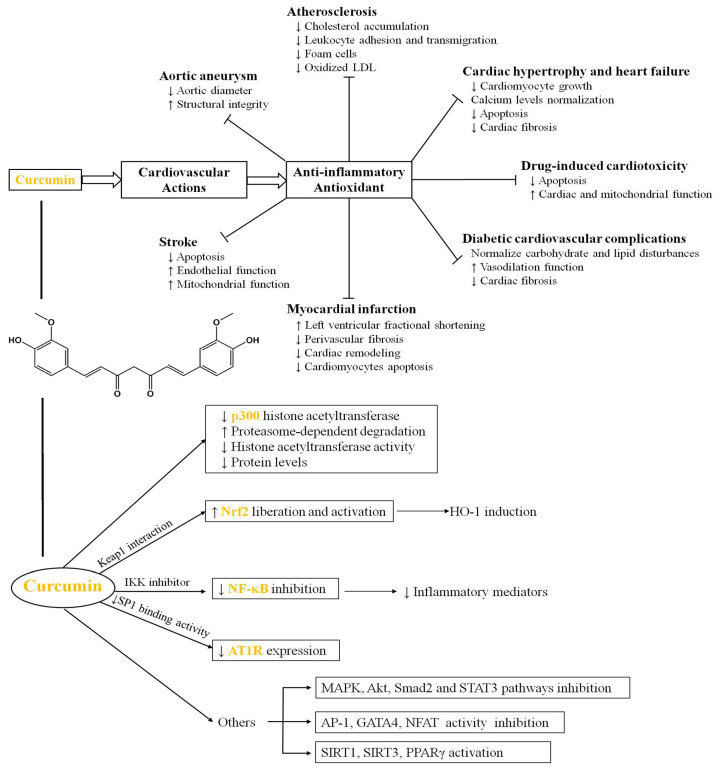
The cardiovascular actions and molecular targets of Cur [61]. Reproduced from Ref. [61] with permission from Elsevier Ltd., copyright 2021.

**Figure 4 pharmaceutics-17-01465-f004:**
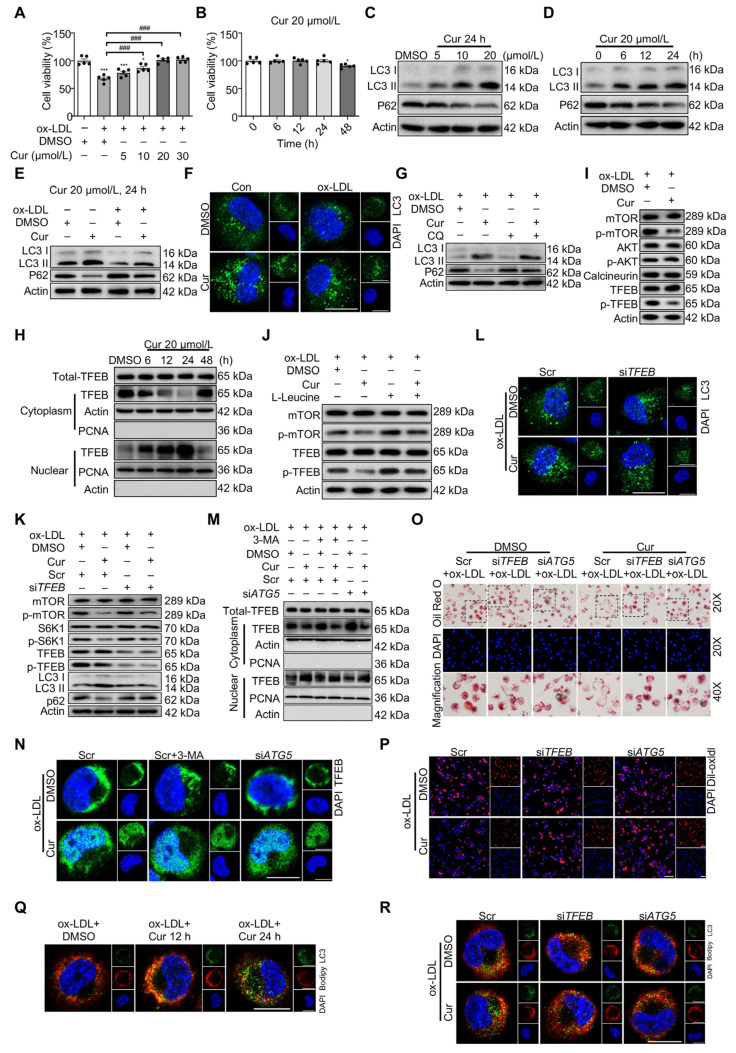
Cur restores FC autophagy and attenuates lipid metabolic dysfunction by promoting TFEB nuclear translocation. (**A**,**B**) Cell viability of FCs was measured using the cell counting kit 8 (CCK-8) assay. (**C**–**E**) Immunoblot analysis of LC3 II and P62 in FCs with different treatments. (**F**) Immunofluorescence analysis of LC3 (green) puncta formation in macrophages treated with Ox-LDL combined with Cur. Scale bar = 20 μm. (**G**) Immunoblot analysis of the autophagic flux in Cur-treated FCs pretreated with chloroquine (CQ). (**H**) Immunoblot analysis of the TFEB subcellular distribution in FCs. (**I**) Immunoblot of mTOR/p-mTOR, AKT/p-AKT, calcineurin, and TFEB/p-TFEB in Cur-treated FCs. (**J**) Immunoblot analysis of the effect of mTORC1-specific agonist l-leucine (0.5 mmol/L) on the activity of mTOR and TFEB in Cur-treated FCs. (**K**,**L**) Immunoblot (**K**) and immunofluorescence (**L**) analysis of autophagy-related proteins in Cur-treated FCs after siTFEB. Scale bar = 20 μm. (**M**,**N**) The subcellular distribution of TFEB in Cur-treated FCs combined with 3-MA and siATG5 was detected using an immunoblot (**M**) and immunofluorescence (**N**). Scale bar = 20 μm. (**O**,**P**) The lipid accumulation (**O**) and the lipid uptake capacity (**P**) of Cur-treated FCs combined with siTFEB or siATG5 were evaluated using oil red O (ORO) staining and 1,1′-dioctadecyl-3,3,3′,3′-tetramethy-lindocarbocyanine perchlorate labeled Ox-LDL (Dil-Ox-LDL), separately. Scale bar = 100 μm. (**Q**) Immunofluorescence analysis of lipid catabolism in FCs conducted using double fluorescence labeling with Bodipy (red) and LC3 (green). Scale bar = 20 μm. (**R**) Immunofluorescence analysis of lipid catabolism in Cur-treated FCs combined with siTFEB or siATG5 through double fluorescence labeling with Bodipy (red) and LC3 (green) at 24 h [71]. Reproduced from Ref. [71] with permission from Springer Ltd., (New York, NY, USA) copyright 2019. * *p* < 0.05, *** *p* < 0.001, ^###^ *p* < 0.001.

**Figure 5 pharmaceutics-17-01465-f005:**
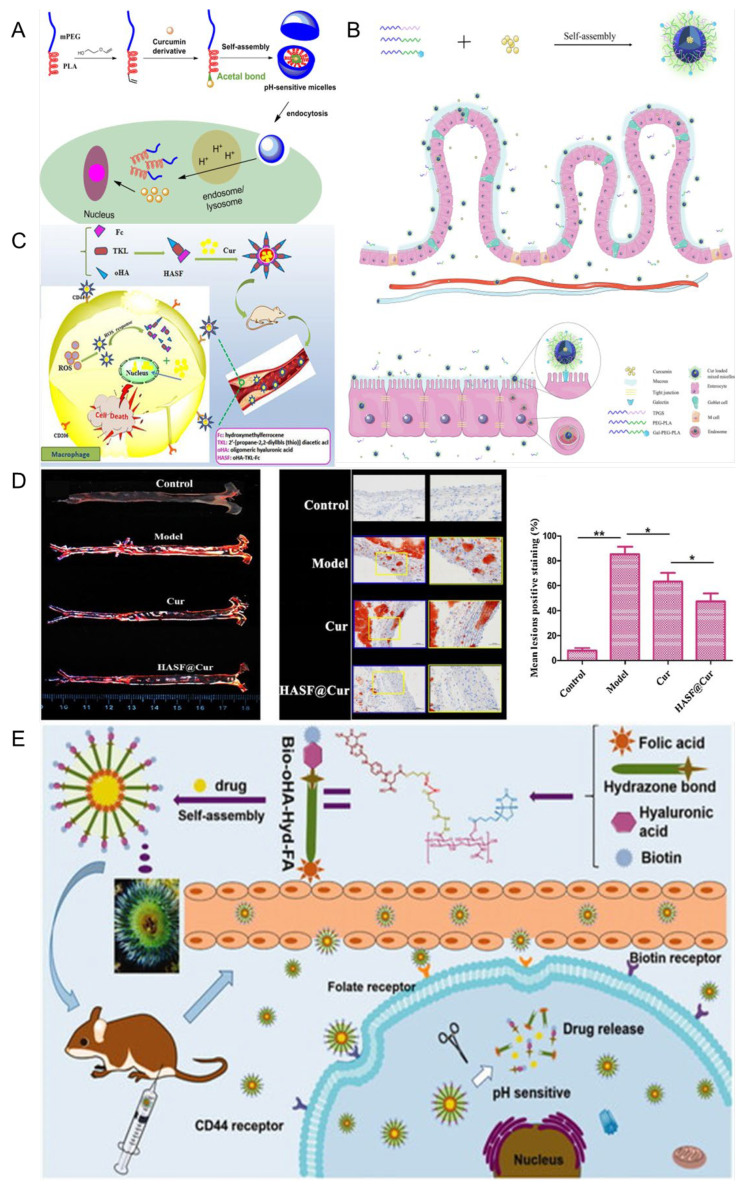
The results of animal experiments using different curcumin formulations. (**A**) An illustration of acetal-linked polymer-Cur conjugate micelle and its endocytosis [114]. Reproduced from Ref. [114] with permission from Elsevier Ltd., copyright 2020. (**B**) The illustration of fabrication and transport of CUR-loaded Gal-PEG-PLA/TPGS micelles [116]. Reproduced from Ref. [116] with permission from Elsevier Ltd., copyright 2015. (**C**) Cur release mechanism diagram of dual ROS-sensitive and CD44 receptors targeting nanoparticles. (**D**) Oil red O lipid staining [117]. Reproduced from Ref. [117] with permission from Elsevier Ltd., copyright 2019. (**E**) AS therapy via ROS-stimulated HASF@Cur micelles in an atherosclerotic rat model [118]. Reproduced from Ref. [118] with permission from Informa Healthcare, copyright 2019. * *p* < 0.05, ** *p* < 0.01.

**Figure 6 pharmaceutics-17-01465-f006:**
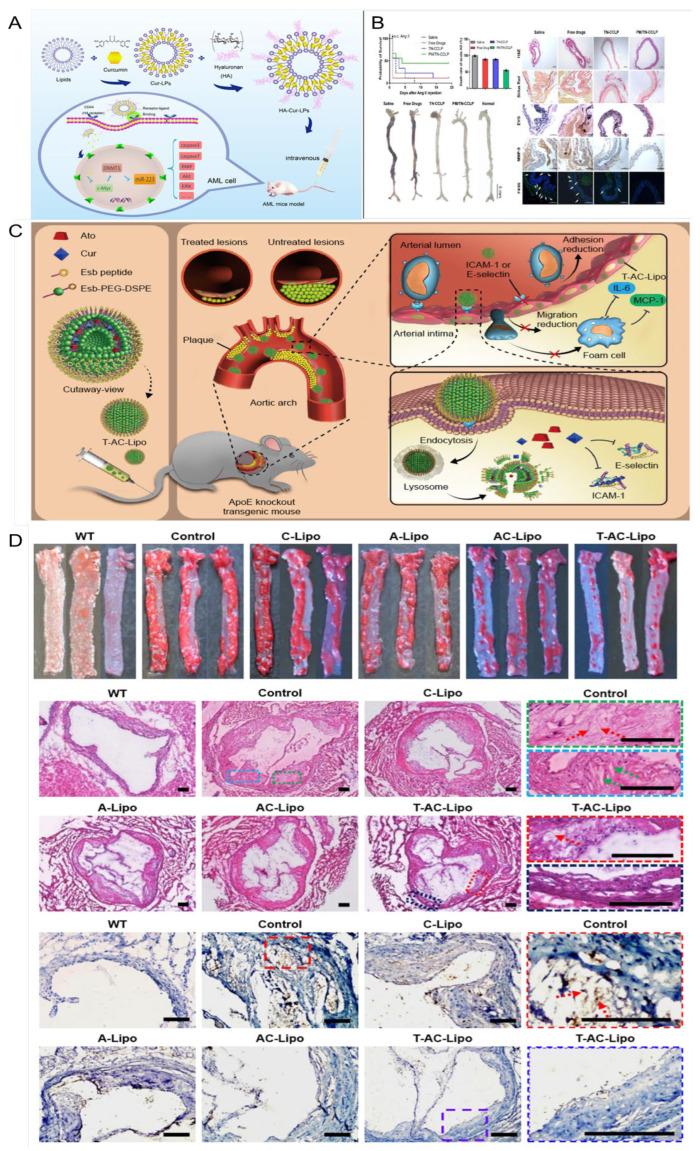
The results of animal experiments using different curcumin formulations. (**A**) Hyaluronic acid-modified Cur liposomal (HA-Cur-LPs) targeted the CD44-specific receptor on the surface of acute myeloid leukemia (AML) cells to deliver Cur to delay AML progression and its mechanism [119]. Reproduced from Ref. [119] with permission from American Chemical Society, copyright 2017. (**B**) In vivo treatment efficacy in the acute AD mice model [121]. Reproduced from Ref. [121] with permission from Elsevier Ltd., copyright 2021. (**C**) Illustration of E-selectin-targeting liposomes (T-AC-Lipo) simultaneously encapsulating Ato and Cur to treat atherosclerotic ApoE^−/−^ mice. (**D**) Oil Red O-stained thoracic aorta and H&E-stained aortic root lesion images for histologic analysis of AS in ApoE^−/−^ mice 30 days after treatment with various liposomes. The dashed green and blue rectangular areas in the control group section and the dashed red and dark blue rectangular areas in the T-AC-Lipo treatment group section are magnified. The dashed red and green arrows indicate foam cells and cholesterol clefts, respectively. [122]. Reproduced from Ref. [122] with permission from Dove Medical Press Ltd., copyright 2019.

**Figure 7 pharmaceutics-17-01465-f007:**
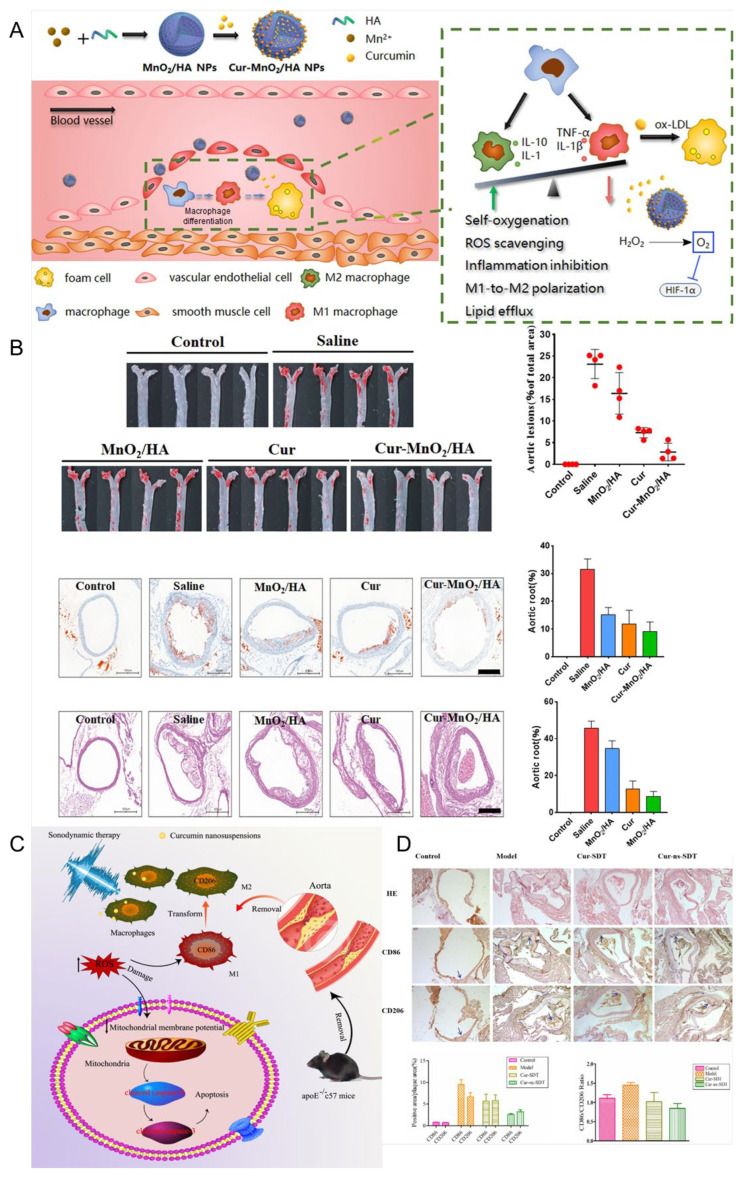
The results of animal experiments using different curcumin formulations. (**A**) Microscopy images of H&E and immunofluorescence analyses (CD 86 and CD 206) in a healthy mouse ankle (Control group), or LPS-infused ankles after treatment with PBS, NPs, free CUR, or NPs-CUR at a CUR dose of 0.25 mg/kg. (**B**) Pictures of atherosclerotic efficacy in mice treated with different groups [127]. Reproduced from Ref. [127] with permission from Elsevier Ltd., copyright 2019. (**C**) A scheme illustrating the preparation of Cur-loaded MnO_2_/HA for targeted delivery to atherosclerotic lesions and the mechanisms of anti-AS therapy. (**D**) The plaque morphology and expression of CD86 and CD206 in ApoE^−/−^ mice were detected via H&E staining and immunohistochemistry staining (*n* = 3). The arrow indicates the positive area. [128]. Reproduced from Ref. [128] with permission from Elsevier Ltd., copyright 2021.

**Figure 8 pharmaceutics-17-01465-f008:**
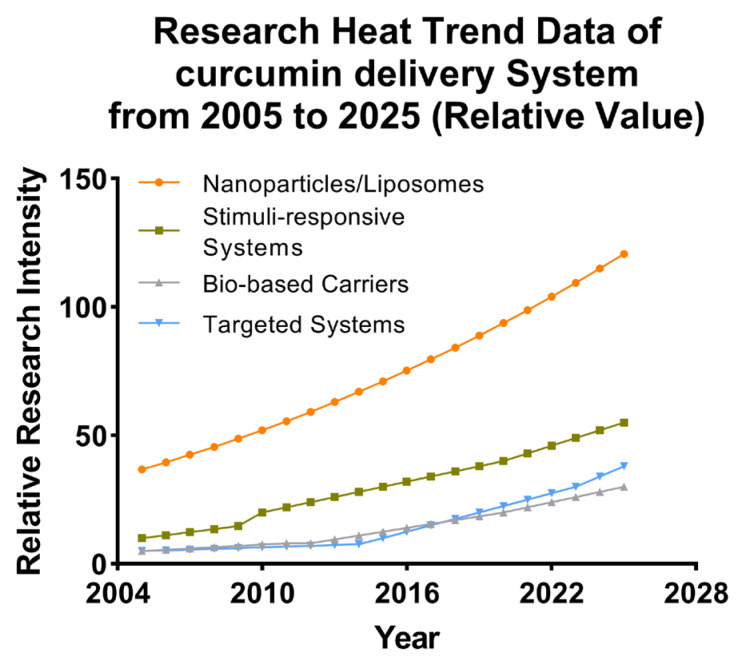
Research trend data for Cur delivery systems from 2005 to 2025 (relative value).

**Table 1 pharmaceutics-17-01465-t001:** A comparison of different nanocarrier systems for Cur delivery.

Delivery System	Advantages	Disadvantages	Therapeutic Efficiency
Polymeric micelles	Small size (~10–100 nm) enables passive targeting via the EPR effect in atherosclerotic plaques. The amphiphilic core–shell structure solubilizes hydrophobic curcumin (solubility ↑ 10–100-fold). Biodegradable polymers (e.g., PLGA and PEG-PLA) reduce systemic toxicity [112].	Low drug-loading efficiency (typically 5–15%). Rapid dissociation in circulation (t_1_/_2_ < 6 h) may require frequent dosing.	Moderate efficiency: passive targeting improves plaque accumulation by 2–3-fold vs. free curcumin; inhibits macrophage foam cell formation [113].
Liposomes	Biocompatible and biodegradable; mimics cell membranes for enhanced cellular uptake. Surface modification (e.g., ligand conjugation) enables active targeting (e.g., CD36 for macrophage targeting). Protects curcumin from enzymatic degradation [59].	Poor physical stability (prone to aggregation/fusion). High production cost; batch-to-batch variability in size distribution.	High efficiency.
Nanoparticles	Versatile materials (e.g., solid lipid nanoparticles and silica NPs) allow for a tunable size (20–200 nm) and sustained release. High drug-loading capacity (up to 30% for lipid-based NPs). Enhanced penetration into atherosclerotic plaques via transcytosis [59].	Potential immunogenicity (e.g., inorganic NPs may trigger cytokine release). Risk of burst release if not surface-modified [141].	High efficiency: sustained release prolongs action (t_1_/_2_ > 24 h); reduces inflammation markers (TNF-α and IL-6) by 40–50%.
Microspheres/ Microcapsules	Sustained release profile (weeks to months) reduces dosing frequency. Macrophage phagocytosis targeting (due to ~1–10 μm size) enhances plaque accumulation.	Their large size limits systemic circulation; cannot cross endothelial barriers efficiently. Their brittle structure may cause premature drug leakage [142].	Low systemic efficiency: localized delivery reduces neointimal hyperplasia by 50% in rabbit models but has limited plaque penetration [132].
Hydrogels	High water content mimics the extracellular matrix; biocompatible for in situ injection. Thermo/pH-responsive release (e.g., PLGA-PEG-PLGA hydrogels) matches the plaque microenvironment (low pH). Protects curcumin from oxidation [135].	Their low degradation rate may lead to incomplete drug release. Poor mechanical strength in dynamic vascular environments [135].	Moderate localized efficiency: in situ injection reduces plaque inflammation by 35% but requires surgical implantation.
Nanosuspensions	Simple preparation (no carrier needed); high drug loading (up to 90%). Small particle size (~200–500 nm) improves oral bioavailability (curcumin solubility ↑ 5–10-fold). Cost-effective for large-scale production [136].	High surface energy causes aggregation in aqueous media. Limited targeting ability; rapid clearance by the reticuloendothelial system [137].	Low efficiency: limited targeting; reduces plasma LDL by 15–20% but shows minimal plaque accumulation [137].

Note: ↑ indicates improvement.

## Data Availability

Data sharing is not applicable to this article, as no new data were created or analyzed in this study.

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
