# Peer review of "Nanostructured Delivery Systems for Curcumin: Improving Bioavailability and Plaque-Targeting Efficacy in Atherosclerosis"

_pharmaceutics, 2025, doi:10.3390/pharmaceutics17111465_

Round 1

Reviewer 1 Report

Comments and Suggestions for Authors

The review on curcumin for treating atherosclerosis is clear and well-structured, covering the disease, curcumin’s effects, and nano-drug delivery systems. To make it stronger, the authors should consider these points:

  1. The authors should include other bioactive compounds and compare them to curcumin.
  2. Authors should mention curcumin’s potential for other conditions (e.g., heart disease, cancer, autoimmune diseases) and explain why the focus is on atherosclerosis.
  3. They should briefly cover alternative or complementary treatments besides curcumin.
  4. Authors should discuss how curcumin can be used in real-world treatments, its long-term safety (e.g., immune responses, long-term effects), and nanomaterial toxicity.
  5. Authors should add quantitative data on curcumin’s effectiveness and how it moves through the body (pharmacokinetics).
  6. They may also think about including a table or section comparing the pros and cons of different nano-carriers.
  7. A section about explaining limitations in current research and issues with delivering curcumin using nanotechnology is needed.
  8. Including a section on research gaps and future directions for balance is also needed.
  9. Curcumin’s mechanism of action in the body is also needed.

Author Response

The review on curcumin for treating atherosclerosis is clear and well-structured, covering the disease, curcumin’s effects, and nano-drug delivery systems. To make it stronger, the authors should consider these points:

  1. The authors should include other bioactive compounds and compare them to curcumin.

Answer: Thank you for your suggestion. Relevant content has been added to Section 4 “Cur in the treatment of AS” in the revised manuscript.

  1. Authors should mention curcumin’s potential for other conditions (e.g., heart disease, cancer, autoimmune diseases) and explain why the focus is on atherosclerosis.

Answer: This is a good question. Curcumin exhibits promising therapeutic potential across diverse conditions—including heart disease, cancer, and autoimmune disorders—primarily attributed to its robust anti-inflammatory and antioxidant properties. However, this study focuses on atherosclerosis because the key pathogenic mechanisms of this disease are oxidative stress and inflammatory responses, and curcumin has a significant role in regulating these mechanisms. Therefore, we hope to explore its mechanism of action in atherosclerosis through this research in depth.

  1. They should briefly cover alternative or complementary treatments besides curcumin.

Answer: We briefly introduce alternative or supplementary treatments other than curcumin, including current surgical interventions and drug treatments for atherosclerosis, in Section 4.

  1. Authors should discuss how curcumin can be used in real-world treatments, its long-term safety (e.g., immune responses, long-term effects), and nanomaterial toxicity.

Answer: We have added a discussion on the potential clinical application of curcumin formulations, emphasizing their real-world therapeutic prospects, long-term safety (including possible immune responses and chronic effects), and nanomaterial-related toxicity concerns. These points have been incorporated into Section 4 and Conclusion in the revised manuscript to provide a more comprehensive outlook on future translational applications.

  1. Authors should add quantitative data on curcumin’s effectiveness and how it moves through the body (pharmacokinetics).

Answer: We have added quantitative data regarding curcumin’s therapeutic efficacy and pharmacokinetic parameters. Specifically, we included results from preclinical and clinical studies reporting on curcumin’s bioavailability in Section 4 in the revised manuscript.

  1. They may also think about including a table or section comparing the pros and cons of different nano-carriers.

Answer: We have added a new table (Table 1) summarizing and comparing the advantages and disadvantages of various nanocarrier systems used for Cur delivery in the revised manuscript.

  1. A section about explaining limitations in current research and issues with delivering curcumin using nanotechnology is needed.

Answer: We have added a new chapter to the manuscript, discussing the current limitations of the research on curcumin and the challenges in delivering curcumin using nanotechnology. Existing research on curcumin, especially in the areas of arteriosclerosis and cardiovascular diseases, is limited by various factors including small sample sizes, short intervention periods, and inconsistent results across different studies. Additionally, a major challenge in curcumin research is its low bioavailability, which has prompted the development of nanotechnology-based delivery systems to enhance its therapeutic potential. However, using nanotechnology to improve the bioavailability of curcumin also brings its own series of challenges. These challenges include the complexity of designing nanoparticles that are stable, biocompatible, have controllable size, surface charge, and release kinetics. Moreover, the long-term safety, potential toxicity, and accumulation in organs such as the liver and spleen also require further research. We have elaborated on this in Section 6 of the revised manuscript.

  1. Including a section on research gaps and future directions for balance is also needed.

Answer: We have added a new chapter to the manuscript, discussing the current limitations of research on curcumin and the challenges in delivering curcumin using nanotechnology in the Conclusion in the revised manuscript.

  1. Curcumin’s mechanism of action in the body is also needed.

Answer: We have discussed the underlying mechanisms of curcumin in treating atherosclerosis, including its anti-inflammatory, lipid-lowering, and anti-thrombotic effects, in Section 4 of the revised manuscript.

Reviewer 2 Report

Comments and Suggestions for Authors

The review comprehensively describes the pathophysiology of atherosclerosis and the possibilities of its treatment.

  1. In the case of the use of curcumin in the therapy of atherosclerosis, it is not clear which data are experimental and which are from in vivo experiments.
  2. It would also be requested to add whether any clinical studies with curcumin have been conducted at all.
  3. When describing the application forms of curcumin, it is only minimally stated that they were tested in the therapy of atherosclerosis.This needs to be added or the title of the article changed, because the title states, various dosage forms of curcumin in the therapy of atherosclerosis and this does not correspond to the main content of the article.
  4. Since the article focuses on curcumin, it is necessary to add the formula of curcumin, not only its chemical name.

Author Response

The review comprehensively describes the pathophysiology of atherosclerosis and the possibilities of its treatment.

  1. In the case of the use of curcumin in the therapy of atherosclerosis, it is not clear which data are experimental and which are from in vivo experiments.

Answer: We are sorry for our lack of clarity. We have carefully revised the manuscript to explicitly distinguish between in vitro and in vivo data. Specifically, the experimental results obtained from in vitro studies and in vivo studies are now clearly indicated in both the main text and figure legends.

  1. It would also be requested to add whether any clinical studies with curcumin have been conducted at all.

Answer: Thank you for your professional suggestion. We have added an overview of clinical studies investigating the use of Cur in AS in Sections 4 and 6 of the revised manuscript. Clinical trials have indeed been conducted to evaluate the effects of Cur on cardiovascular risk factors such as lipid profiles, blood glucose, and inflammatory markers. However, the overall clinical evidence is still limited, and more extensive trials are needed to confirm the therapeutic potential of Cur in atherosclerosis and other cardiovascular conditions.

  1. When describing the application forms of curcumin, it is only minimally stated that they were tested in the therapy of atherosclerosis. This needs to be added or the title of the article changed, because the title states, various dosage forms of curcumin in the therapy of atherosclerosis and this does not correspond to the main content of the article.

Answer: Thank you for your suggestion. We have modified the title in the revised manuscript.

  1. Since the article focuses on curcumin, it is necessary to add the formula of curcumin, not only its chemical name.

Answer: This has been added in the Section 4 and Figure 3 in the revised manuscript.

Reviewer 3 Report

Comments and Suggestions for Authors
  1. The review currently reads more like a compilation of existing studies than a critical synthesis. The authors should include a comparative analysis of results from different studies, highlight contradictory findings, and discuss the limitations and challenges of current research on curcumin in AS.
  2. The manuscript provides extensive mechanistic detail (e.g., NF-őļB, MAPK, PI3K-Akt pathways) but often lacks explanation of how these pathways interact or differ in vivo.
  3. Several statements (especially regarding nanotechnology applications) lack direct supporting citations.
  4. Several sections (e.g., Sections 3 and 4) overlap in content, particularly regarding lipid-lowering and anti-inflammatory mechanisms. Consider using subheadings for "Mechanisms of Action" and "Delivery Strategies."
  5. Although numerous figures are referenced, it is unclear whether all are original or adapted. Figure legends should specify the source and provide concise explanatory captions.
  6. The manuscript contains frequent grammatical errors, awkward phrasing, and inconsistent terminology (e.g., "Cur" vs. "curcumin"). Terms such as "Turmeric, also known as turmeric" should be corrected, and redundant sentences removed for conciseness.
  7. The conclusion restates content rather than providing insight or identifying research gaps. Strengthen the conclusion by discussing challenges for clinical translation, regulatory considerations, and potential strategies to improve curcumin's pharmacokinetic profile.

Author Response

The review currently reads more like a compilation of existing studies than a critical synthesis. The authors should include a comparative analysis of results from different studies, highlight contradictory findings, and discuss the limitations and challenges of current research on curcumin in AS.

  1. The manuscript provides extensive mechanistic detail (e.g., NF-őļB, MAPK, PI3K-Akt pathways) but often lacks explanation of how these pathways interact or differ in vivo.

Answer: We have expanded the discussion in Section 4.1 of the revised manuscript. Specifically, we described how curcumin may modulate multiple inflammatory and oxidative stress-related cascades simultaneously, and how the cross-talk between these pathways contributes to the regulation of macrophage polarization, endothelial protection, and lipid metabolism during atherosclerosis progression.

  1. Several statements (especially regarding nanotechnology applications) lack direct supporting citations.

Answer: Thank you for your suggestion. We have carefully reviewed the manuscript and added additional, more recent references to support the statements, especially those related to nanotechnology applications in the revised manuscript. These citations strengthen the scientific basis of our discussion.

  1. Several sections (e.g., Sections 3 and 4) overlap in content, particularly regarding lipid-lowering and anti-inflammatory mechanisms. Consider using subheadings for "Mechanisms of Action" and "Delivery Strategies."

Answer: We have clarified the structure and focus of Sections 3 and 4 to avoid content overlap in the revised manuscript. Specifically, Section 3 now primarily discusses the current clinically used drugs and therapeutic strategies for atherosclerosis, providing a pharmacological background. In contrast, Section 4 focuses on the therapeutic effects and mechanisms of curcumin in atherosclerosis, including its lipid-lowering, anti-inflammatory, and antioxidant activities. While some mechanistic aspects inevitably overlap (as both conventional drugs and curcumin target similar inflammatory and lipid pathways), we have revised the text to minimize redundancy.

  1. Although numerous figures are referenced, it is unclear whether all are original or adapted. Figure legends should specify the source and provide concise explanatory captions.

Answer: Thank you for your suggestion. We have revised all figure legends to specify whether each figure is original or adapted, and added concise explanatory captions to improve readability in the revised manuscript.

  1. The manuscript contains frequent grammatical errors, awkward phrasing, and inconsistent terminology (e.g., "Cur" vs. "curcumin"). Terms such as "Turmeric, also known as turmeric" should be corrected, and redundant sentences removed for conciseness.

Answer: Thanks for your thoughtful suggestions. Our manuscript was polished by the editing agency recommended by the journal.

  1. The conclusion restates content rather than providing insight or identifying research gaps. Strengthen the conclusion by discussing challenges for clinical translation, regulatory considerations, and potential strategies to improve curcumin's pharmacokinetic profile.

Answer: Thank you for your professional suggestion. The Conclusion has been substantially expanded to provide deeper insights rather than simply restating the content. We have added a discussion on the challenges in clinical translation, regulatory considerations, and potential strategies to improve curcumin’s pharmacokinetic profile, such as nanoformulation optimization and structural modification. These revisions aim to highlight future research directions and the practical significance of curcumin-based therapies.

Reviewer 4 Report

Comments and Suggestions for Authors

The review submitted by Liu et al. discuss the latest research papers concerning the use of curcumin for the treatment of atherosclerosis. The manuscript is clear, well written and structured. However, there are some point that can be improved:

  1. the authors can indicate the keywords and the databases used for their research.
  2. the resolution of fig 3 must be enhanced.
  3. the hydrogels based on polysaccharides must be better discussed
  4.  curcumin was also encapsulated in electrospun fibers and in other systems and therefore a section regarding "miscellaneous systems" could be of interest
  5. a section of critical personal remarks must be added
  6. more perspectives could also be proposed by the authors. 

Author Response

The review submitted by Liu et al. discuss the latest research papers concerning the use of curcumin for the treatment of atherosclerosis. The manuscript is clear, well written and structured. However, there are some point that can be improved:

  1. the authors can indicate the keywords and the databases used for their research.

Answer: The keywords and databases used in our research have been clearly indicated in the revised manuscript.
Keywords: atherosclerosis; curcumin; inflammation; nanoformulations; drug delivery. We mainly searched the literature from Elsevier, Clarivate Analytics, Springer Nature, and Wiley.

  1. the resolution of fig 3 must be enhanced

Answer: It has been modified in the revised manuscript.

  1. the hydrogels based on polysaccharides must be better discussed

Answer: We have expanded the discussion on polysaccharide-based hydrogels, and examples have been added to the revised manuscript.

  1. curcumin was also encapsulated in electrospun fibers and in other systems and therefore a section regarding "miscellaneous systems" could be of interest

Answer: We have added a new subsection in the revised manuscript. This section now includes recent advances in electrospun fiber-based delivery systems, which have shown great potential in providing localized, sustained release of curcumin for cardiovascular repair and tissue regeneration.

  1. a section of critical personal remarks must be added

 Answer: We have added a new chapter to the manuscript, discussing the limitations of existing research on curcumin and the challenges in delivering curcumin using nanotechnology. In particular, we have elaborated on this in Section 6 of the revised manuscript.

  1. more perspectives could also be proposed by the authors. 

Answer: We have added a new section to the revised manuscript, aiming to further elaborate on the future development of the curcumin delivery system. The section focuses on introducing the new strategies and the challenges encountered during the transformation process.

Reviewer 5 Report

Comments and Suggestions for Authors

1. The review is written as a list of facts, without summarizing the conclusions, etc. It's assumed that each point or subpoint in the review serves a purpose, and that each point or subpoint should end with some kind of conclusion. This isn't the case...
2. The review is, at best, no better than reviews 10.3390/molecules26134036, 10.25163/angiotherapy.839555, 10.1155/2020/1520747, 10.1093/nutrit/nuaf012, 10.1002/biof.1603, and several dozen others. Moreover, the authors cite a single review, as if pretending the topic hasn't been addressed before (L. Zeng, T. Yang, K. Yang, G. Yu, J. Li, W. Xiang, H. Chen, Curcumin and Curcuma longa Extract in the Treatment of 10 Types of Autoimmune Diseases: A Systematic Review and Meta-Analysis of 31 Randomized Controlled Trials, Front Immunol 13 831(2022) 896476.)
3. The review focuses on different dosage forms, yet the authors didn't construct any graphs showing the efficacy or even the degree of influence of the different forms. This is odd, as other reviews provide this information.
4. The review doesn't contain any original figures. All figures are borrowed from well-known experimental articles. I believe this is only acceptable at the very beginning of a topic's development, when information is still scant. Considering that reviews on the topic have been written for almost two decades, this is very odd. Didn't the authors find any worthy facts to illustrate?
5. Most of the other reviews contain diagrams of the molecular mechanisms. This is quite important and in demand among readers. My cursory examination of the field revealed that this is precisely where significant progress has been made by biologists and physicians in the past few years.
6. I didn't find a single reference in the review for the year 2025. This is alarming! Only one reference for 2024. Therefore, I suggest the authors change the title of the review. The new title should be: "Application of curcumin in different dosage forms in the treatment of atherosclerosis (data for 2023 and later)".
7. The manuscript doesn't clearly explain why traditional clinically proven treatments are inferior to curcumin. This is likely one of the main questions readers will be asking themselves. A clear comparison is needed! Incidentally, this is also mentioned in previously published reviews. The authors' opinion is essentially expressed in the conclusion, stating that traditional anti-atherosclerosis medications demonstrate efficacy.
8. The conclusion is contradictory. On the one hand, the authors write: "Cur emerges as a crucial natural therapeutic agent for AS treatment." On the other, they write: "However, due to its low solubility and poor stability, modifications to its dosage form and mode of administration are necessary to enhance its pharmacological efficacy." In short, it's complicated...

It's too early to publish the review's conclusion in its current form; it needs to be corrected and improved.

Author Response

  1. The review is written as a list of facts, without summarizing the conclusions, etc. It's assumed that each point or subpoint in the review serves a purpose, and that each point or subpoint should end with some kind of conclusion. This isn't the case...

Answer: We have added two new sections in the revised version, further discussing the existing limitations of curcumin research and the challenges faced by the delivery systems based on nanotechnology (Section 6). Currently, the research on curcumin in the field of atherosclerosis and cardiovascular diseases is still limited by factors such as small sample size, short intervention period, and inconsistent research results. Its low bioavailability is also a major obstacle. Therefore, various nanotechnology delivery systems have been developed to enhance its therapeutic effect. However, the application of nanotechnology has also brought new challenges, such as the complexity of carrier design (stability, biocompatibility, particle size, surface charge, and controllability of drug release behavior), as well as potential long-term safety, toxicity, and organ accumulation issues. Furthermore, in the revised version, we added a section on the future development prospects of curcumin delivery systems, focusing on new strategies such as stimulus-responsive, composite, and biomimetic nanocarriers. We also discussed the challenges encountered during the clinical translation process, including large-scale production, regulatory approval, and long-term safety evaluation. Through these additions, we aim to make the discussion more systematic and forward-looking.

  1. The review is, at best, no better than reviews 10.3390/molecules26134036, 10.25163/angiotherapy.839555, 10.1155/2020/1520747, 10.1093/nutrit/nuaf012, 10.1002/biof.1603, and several dozen others. Moreover, the authors cite a single review, as if pretending the topic hasn't been addressed before (L. Zeng, T. Yang, K. Yang, G. Yu, J. Li, W. Xiang, H. Chen, Curcumin and Curcuma longa Extract in the Treatment of 10 Types of Autoimmune Diseases: A Systematic Review and Meta-Analysis of 31 Randomized Controlled Trials, Front Immunol 13 831(2022) 896476.)

Answer: We have sought to address the reviewer's concern by clarifying the review's unique focus on curcumin nanodelivery systems for atherosclerosis (vs. general pharmacology in cited studies), supplementing some recent references, and highlighting novel analyses of nanocarrier and clinical translation data. We also changed the title to make it more appropriate in the revised manuscript.

  1. The review focuses on different dosage forms, yet the authors didn't construct any graphs showing the efficacy or even the degree of influence of the different forms. This is odd, as other reviews provide this information.

Answer: We have added a new table (Table 1) summarizing and comparing the advantages and disadvantages of various nanocarrier systems used for Cur delivery in the revised manuscript.

  1. The review doesn't contain any original figures. All figures are borrowed from well-known experimental articles. I believe this is only acceptable at the very beginning of a topic's development, when information is still scant. Considering that reviews on the topic have been written for almost two decades, this is very odd. Didn't the authors find any worthy facts to illustrate?

Answer: Thank you for your professional suggestion. Table 1 and Figure 8 have been added to the revised manuscript.

  1. Most of the other reviews contain diagrams of the molecular mechanisms. This is quite important and in demand among readers. My cursory examination of the field revealed that this is precisely where significant progress has been made by biologists and physicians in the past few years.

Answer: Figure 1 and Figure 3 were modified in the revised manuscript, which include a schematic representation of the molecular mechanisms of curcumin intervention based on the pathological progression of atherosclerosis. The updated figure now highlights curcumin’s multi-target effects, including its anti-inflammatory, antioxidant, lipid-regulating, and plaque-stabilizing actions.

  1. I didn't find a single reference in the review for the year 2025. This is alarming! Only one reference for 2024. Therefore, I suggest the authors change the title of the review. The new title should be: "Application of curcumin in different dosage forms in the treatment of atherosclerosis (data for 2023 and later)".

Answer: This is a good suggestion. We have carefully updated the references throughout the manuscript to include the most recent studies published in 2024 and 2025, ensuring that the review reflects the latest progress in this research field.

  1. The manuscript doesn't clearly explain why traditional clinically proven treatments are inferior to curcumin. This is likely one of the main questions readers will be asking themselves. A clear comparison is needed! Incidentally, this is also mentioned in previously published reviews. The authors' opinion is essentially expressed in the conclusion, stating that traditional anti-atherosclerosis medications demonstrate efficacy.

Answer: In the revised manuscript, we have added a comparative discussion on traditional anti-atherosclerotic therapies and treatments based on curcumin to clarify their respective advantages and limitations. Specifically, we emphasize that traditional clinical drugs, such as statins and anti-inflammatory drugs, have clear efficacy in reducing lipid levels and preventing cardiovascular events, but their long-term use may cause side effects such as liver damage, muscle disorders, and drug resistance. In contrast, curcumin, as a natural polyphenolic compound, has multiple benefits such as anti-inflammatory, antioxidant, regulation of lipid levels, and protection of the endothelium, and is relatively safe. However, its clinical application is limited by low bioavailability and rapid metabolism, which requires the development of advanced nano-delivery systems to improve its pharmacokinetics and therapeutic performance.

  1. The conclusion is contradictory. On the one hand, the authors write: "Cur emerges as a crucial natural therapeutic agent for AS treatment." On the other, they write: "However, due to its low solubility and poor stability, modifications to its dosage form and mode of administration are necessary to enhance its pharmacological efficacy." In short, it's complicated... It's too early to publish the review's conclusion in its current form; it needs to be corrected and improved.

Answer: Thank you for your professional suggestion. In the revised manuscript, we have rephrased this section to clarify that while curcumin (Cur) possesses significant therapeutic potential against atherosclerosis (AS) due to its multiple pharmacological effects, its clinical translation remains limited by poor solubility and stability. Therefore, curcumin itself is a promising natural agent, but its efficacy can be further enhanced through optimized formulations and delivery systems.

Round 2

Reviewer 1 Report

Comments and Suggestions for Authors

The authors have responded properly to most of the comments and the manuscript has been significantly improved and can be published

Author Response

The authors have responded properly to most of the comments and the manuscript has been significantly improved and can be published.

Answer: Thank you for your comments. Our manuscript was polished by the editing agency recommended by the journal.

Reviewer 2 Report

Comments and Suggestions for Authors

The authors accepted all recomendations and improved the manuscript..

Author Response

The authors accepted all recommendations and improved the manuscript.

Answer: Thank you for your comments.

Reviewer 3 Report

Comments and Suggestions for Authors

Accept in present form

Author Response

Accept in present form

Answer: Thank you for your comments.

Reviewer 5 Report

Comments and Suggestions for Authors

The authors revised the conclusion—that's good.
The authors compared nano and traditional methods—that's also good.
The authors read the articles from 2025 and 2024 and added relevant information to the review—that's excellent.
The authors, albeit weakly, examined the molecular mechanisms—that's good.
The authors changed the title, and now it more accurately reflects the content—that's excellent.
Despite all the apparent success, two key points remain that haven't even begun to be corrected. In my opinion, the review is a jumble of facts that are weakly interconnected. Personally, I don't always understand why the authors mention certain facts. Ideally, each paragraph of text should end with a conclusion. I think it would be a good idea in this case for the authors to draw conclusions after the subsections and sections. Conclusions allow the information to be conveyed from the authors to the reader in a succinct manner. The second thing that bothers me is the lack (now incomplete) of author's images. Why do they even need such images? You can simply link to the article and the image number. Often, images from experimental studies focus on a narrow aspect of the phenomenon being studied. This is why, after reading of manuscript one might conclude that most of the images illustrate something unrelated or incomplete...

Author Response

The authors revised the conclusion—that's good.

The authors compared nano and traditional methods—that's also good.

The authors read the articles from 2025 and 2024 and added relevant information to the review-that's excellent.
The authors, albeit weakly, examined the molecular mechanisms—that's good.
The authors changed the title, and now it more accurately reflects the content—that's excellent.
Despite all the apparent success, two key points remain that haven't even begun to be corrected. In my opinion, the review is a jumble of facts that are weakly interconnected. Personally, I don't always understand why the authors mention certain facts. Ideally, each paragraph of text should end with a conclusion. I think it would be a good idea in this case for the authors to draw conclusions after the subsections and sections. Conclusions allow the information to be conveyed from the authors to the reader in a succinct manner. The second thing that bothers me is the lack (now incomplete) of author's images. Why do they even need such images? You can simply link to the article and the image number. Often, images from experimental studies focus on a narrow aspect of the phenomenon being studied. This is why, after reading of manuscript one might conclude that most of the images illustrate something unrelated or incomplete...

Answer: Thank you for your valuable comments. You specifically raised two points regarding our manuscript, and we have carefully revised the manuscript accordingly. First, we have provided comprehensive discussions and summaries for each section in the revised version. Second, we have added detailed interpretive analyses of the figures introduced in our review, clarifying their implications and reinforcing their connection to the main arguments.

Round 3

Reviewer 5 Report

Comments and Suggestions for Authors

The quality of the manuscript has become acceptable for publication